# Rapid adaptation of endocytosis, exocytosis, and eisosomes after an acute increase in membrane tension in yeast cells

**Joël Lemière[1,2†‡], Yuan Ren[1,2†], Julien Berro[1,2,3]\***

[1]Department of Molecular Biophysics and Biochemistry, Nanobiology Institute, Yale University, New Haven, United States; [2]Nanobiology Institute, Yale University, West Haven, United States; [3]Department of Cell Biology, Yale University School of Medicine, New Haven, United States

**Abstract** During clathrin-mediated endocytosis (CME) in eukaryotes, actin assembly is required to overcome large membrane tension and turgor pressure. However, the molecular mechanisms by which the actin machinery adapts to varying membrane tension remain unknown. In addition, how cells reduce their membrane tension when they are challenged by hypotonic shocks remains unclear. We used quantitative microscopy to demonstrate that cells rapidly reduce their membrane tension using three parallel mechanisms. In addition to using their cell wall for mechanical protection, yeast cells disassemble eisosomes to buffer moderate changes in membrane tension on a minute time scale. Meanwhile, a temporary reduction in the rate of endocytosis for 2–6 min and an increase in the rate of exocytosis for at least 5 min allow cells to add large pools of membrane to the plasma membrane. We built on these results to submit the cells to abrupt increases in membrane tension and determine that the endocytic actin machinery of fission yeast cells rapidly adapts to perform CME. Our study sheds light on the tight connection between membrane tension regulation, endocytosis, and exocytosis.

**\*For correspondence:**
julien.berro@yale.edu

[†]These authors contributed equally to this work

**Present address:** [‡]UCSF, Department of Cell and Tissue Biology, San Francisco, United States

**Competing interests:** The authors declare that no competing interests exist.

## Introduction

During clathrin-mediated endocytosis (CME), the cell plasma membrane undergoes a dramatic change in topology to form an invagination that is subsequently pinched off into a vesicle. During this process, the endocytic machinery has to overcome the forces produced by membrane tension and the osmotic pressure that opposes membrane deformation and engulfment. In yeast cells, these resisting forces are particularly large because their internal turgor pressure is high, ranging from ~0.6 MPa for *Saccharomyces cerevisiae* to more than 1 MPa for *Schizosaccharomyces pombe* (*Davì et al., 2018*; *Minc et al., 2009*; *Schaber et al., 2010*). Consequently, the formation of a vesicle requires several thousands of piconewtons (pN) (*Dmitrieff and Nédélec, 2015*; *Ma and Berro, 2021*).

Previous studies have shown that actin dynamics is required for productive endocytosis in yeast (*Aghamohammadzadeh et al., 2014*; *Basu et al., 2014*; *Carlsson and Bayly, 2014*; *Palmer et al., 2015*) and in mammalian cells when membrane tension is high (*Aghamohammadzadeh and Ayscough, 2009*; *Boulant et al., 2011*; *Hassinger et al., 2017*), or when membrane scission proteins are absent (*Ferguson et al., 2009*). Actin assembly at the endocytic site is believed to provide the forces that overcome turgor pressure and membrane tension to deform the plasma membrane, but the precise mechanisms of force production remain unknown (reviewed in *Berro and Lacy, 2018*; *Goode et al., 2015*; *Lacy et al., 2018*). We also lack a quantitative understanding of the

regulation of actin dynamics in response to membrane tension and turgor pressure changes. We expect that a better quantitative characterization of this response will allow us to infer the molecular mechanisms of force production and force sensing during CME.

The mechanisms by which membrane tension is regulated are not fully understood. The yeast cell wall is believed to buffer abrupt changes in turgor pressure, thanks to its high stiffness of ~50 MPa (*Atilgan et al., 2015*). In addition, similarly to mammalian cells' caveolae, which change shape or disassemble in response to increased membrane tension, yeast eisosomes can also disassemble when cells without a cell wall, called protoplasts, are placed in low osmolarity media (*Kabeche et al., 2015*; *Parton et al., 2020*; *Sinha et al., 2011*). However, it remains unknown how eisosomes may regulate plasma membrane tension in intact cells and whether eisosome disassembly directly influences cellular processes such as CME. In addition, it remains unclear to which extent endocytosis and exocytosis may contribute to the regulation of membrane tension when the cells are challenged with an abrupt increase in membrane tension.

Fission yeast is an ideal model system to quantitatively study the regulation mechanisms of membrane tension and its influence on the endocytic machinery. First, because yeast turgor pressure is high, actin is required for CME. Second, contrary to mammalian cells, yeast cells are devoid of any adhesion machinery or actin cortex, which usually complicates membrane tension manipulation and result interpretation. Last, quantitative microscopy methods developed in fission yeast are able to uncover fine regulations of the endocytic machinery and precisely measure the local and global numbers of endocytic events at a given time (*Arasada and Pollard, 2011*; *Berro et al., 2010*; *Berro and Lacy, 2018*; *Berro and Pollard, 2014a*; *Berro and Pollard, 2014b*; *Chen and Pollard, 2013*; *Lacy et al., 2019*; *Sirotkin et al., 2010*).

To probe the contributions of each possible mechanism of membrane tension regulation and their influence on CME, we submitted yeast cells with or without a cell wall to different hypotonic shocks. Using quantitative fluorescence microscopy, we showed that, on one hand, yeast cells rapidly reduce their membrane tension by (a) disassembling eisosomes, (b) reducing their rate of endocytosis, and (c) increasing their rate of exocytosis and, on the other hand, actin assembly adapts to increased membrane tension to allow endocytosis to proceed.

## Results

### Eisosomes participate in the regulation of protoplasts' membrane tension

Previous studies have proposed that eisosomes, furrows at the inner surface of the plasma membrane, have a mechanoprotective role under increased membrane tension in fungi by acting as a reservoir of membrane, similar to the protective role of caveolae in endothelial cells (*Cheng et al., 2015*; *Kabeche et al., 2015*; *Lo et al., 2016*; *Sens and Turner, 2006*; *Sinha et al., 2011*). Since the yeast cell wall plays a major role in the maintenance of cell integrity under extreme osmotic conditions, thanks to its high stiffness of ~50 MPa (*Atilgan et al., 2015*), it prevents large variations in membrane tension under hypotonic shocks. Hence, to exclude the effect of the cell wall and amplify membrane tension changes, we performed our experiments using cells devoid of a cell wall, hereafter referred to as 'protoplasts', instead of intact cells, hereafter referred to as 'walled cells'.

First, we characterized how the removal of the cell wall affects eisosomes' reorganization. We used a protocol that allowed us to manipulate protoplasts for up to ~1 hr after their formation, since they remain void of cell wall for about 3 hr (*Flor-Parra et al., 2014*). Because protoplasts are more fragile than walled cells, they were prepared in Edinburgh Minimum Media (EMM5S) containing 0.25–1.2 M sorbitol to balance turgor pressure and prevent cells from bursting, while keeping nutrient concentration constant (*Basu et al., 2014*; *Kabeche et al., 2015*; *Stachowiak et al., 2014*), and were imaged ~15 min later, once they reached steady state. In the rest of the paper, we will refer to this experimental condition as 'steady state in X M' or 'chronic exposure to X M sorbitol', where X is the sorbitol concentration.

Our data show that eisosomes in protoplasts at steady state in 1.2 M sorbitol are qualitatively similar to those in walled cells (*Figure 1A*), and the cellular concentration of Pi1lp is the same in both conditions (*Figure 1E*). However, the surface area of the protoplasts' plasma membrane covered by eisosomes decreased with decreasing media osmolarity at steady state (*Figure 1B and C*)

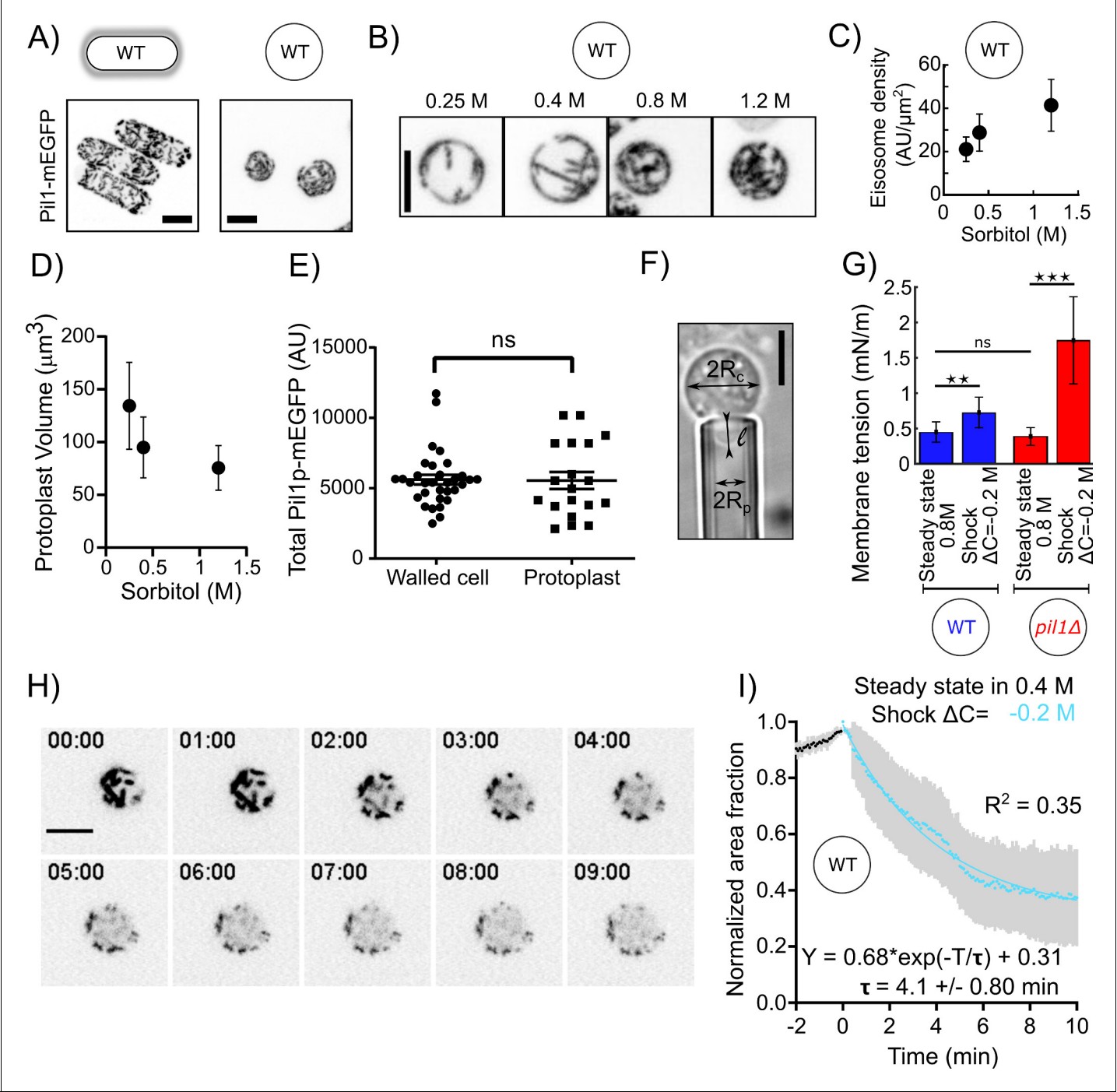

**Figure 1.** Eisosomes disassemble to buffer increases in membrane tension. (A) Representative walled yeast cells (left column) and protoplasts (right column) at steady state in 1.2 M sorbitol expressing eisosome core protein Pil1-mEGFP (inverted contrast). Note that the cellular concentration of Pil1p-mEGFP is the same in walled cells and protoplasts (panel E). (B) Eisosomes labeled with Pil1p-mEGFP (inverted contrast) in wild-type protoplasts at steady state in different sorbitol concentrations. From left to right: 0.25, 0.4, 0.8, and 1.2 M sorbitol. (C and D) Density of eisosomes at the plasma membrane (C), measured as the ratio between the intensity of Pil1p-mEGFP on the plasma membrane and the surface area of the protoplast and volume (D), at steady state in 0.25 M (N = 26), 0.4 M (N = 34), and 1.2 M (N = 39) sorbitol. Error bars: standard deviations. (E) The total amount of Pil1-mEGFP in walled cells (N = 32) and protoplasts in 0.4 M sorbitol (N = 19) are not significantly different (Mann-Whitney test, p=0.65). Each point represents one measurement; bars are the mean and SEM. (F) Micropipette aspiration was used to measure membrane tension. $R_c$: cell radius; $R_p$: micropipette radius; l: length of the tongue inside the micropipette. (G) Membrane tension of protoplasts at steady state in 0.8 M sorbitol and ~5 min after a hypotonic shock (ΔC = −0.2 M) for wild-type (blue bars, N = 28 for steady state and N = 5 for the shock) and *pil1Δ* protoplasts (red bars, N = 42 for steady state and N = 7 for the shock). Error bars: standard deviation. p-values: non-significant (ns), p>0.05; **p≤0.01; ***p≤0.001. (H and I)

*Figure 1 continued on next page*

*Figure 1 continued*

Eisosomes of wild-type protoplasts disassemble rapidly after a hypotonic shock. (H) Time course of a representative protoplast expressing Pil1p-mEGFP over 10 min after a hypotonic shock (ΔC = −0.2 M) and initially at steady state in 0.4 M sorbitol (just before time 0 min). (I) Evolution of the surface area covered by eisosomes over time, as a fraction of the surface area covered at time 0 min (normalized to 1). Data are from three independent experiments (N = 15) and presented as mean ± 95% confidence interval. Scale bars in (A), (B), (F), and (H): 5 μm.

The online version of this article includes the following source data for figure 1:

**Source data 1.** Data for *Figure 1C* and 1D.
**Source data 2.** Data for *Figure 1E*.
**Source data 3.** Data for *Figure 1G*.
**Source data 4.** Data for *Figure 1I*.

and correlated with increasing cell volume (*Figure 1D*). This result confirms previous results (*Kabeche et al., 2015*) showing that eisosomes are disassembled in media with low osmolarity and the disassembly of eisosomes may reduce membrane tension.

We then performed hypotonic shocks to abruptly increase the membrane tension of protoplasts and we imaged them before their long-term adaptation to changes in media osmolarity. Prior to the shocks, we let cells reach steady state by exposing them to media with a given sorbitol concentration for more than 15 min. We chose 0.4 M sorbitol as the steady-state concentration for this experiment as it corresponds to the estimated turgor pressure of walled cells (~1 MPa), and at this concentration, the dynamics of the actin endocytic machinery was virtually identical to the one in walled cells (*Figure 7—figure supplements 1B*). We performed acute hypotonic shocks by using a microfluidic system to rapidly exchange the steady state media with media containing a lower sorbitol concentration, hereafter noted as $\Delta C = -$ Y M, where Y is the difference in media osmolarity (note that the change in pressure $\Delta P$ in Pascal is related to the change in osmolite concentration $\Delta C$ in Molar as $\Delta P = \Delta C \cdot RT \sim 2.45 \cdot 10^6 \cdot \Delta C$, where $R$ is the gas constant and $T$, the absolute temperature, when the solute concentration is sufficiently low).

To quantitatively characterize eisosome disassembly in protoplasts after a hypotonic treatment, we measured the temporal evolution of the decrease in surface area covered by eisosomes after an acute hypotonic shock of $\Delta C = -0.2$ M starting with protoplasts at steady state in 0.4 M sorbitol (*Figure 1H and I*). Eisosomes disassembled rapidly after the hypotonic shock, dropping to ~50% of the surface area covered by eisosomes before the shock within 5 min, indicating a fast response to counteract the hypotonic shock and an eventual change in membrane tension.

To test whether membrane tension is buffered by eisosomes, we measured membrane tension using a micropipette aspiration assay (*Figure 1F*). At steady state in 0.8 M sorbitol, the membrane tension was $0.45 \pm 0.14$ mN·m$^{-1}$ for wild-type (WT) protoplasts and $0.39 \pm 0.13$ mN·m$^{-1}$ for *pil1Δ* protoplasts (*Figure 1G*). We then repeated these measurements within 5 min after inducing a hypotonic shock of $\Delta C = -0.2$ M. We observed a 1.6-fold increase in membrane tension for WT protoplasts ($0.73 \pm 0.21$ mN·m$^{-1}$) and a 4.5-fold increase for protoplasts lacking eisosomes ($1.74 \pm 0.61$ mN·m$^{-1}$). This result demonstrates that eisosomes participate in the adjustment of plasma membrane tension.

## In protoplasts, eisosomes buffer moderate hypotonic shocks

Since WT protoplasts were able to withstand osmotic shocks by disassembling their eisosomes, we hypothesized that protoplasts lacking eisosomes – either because they lack the core eisosome protein Pil1p or because eisosomes are mechanically removed – are more sensitive to hypotonic shocks.

WT and *pil1Δ* protoplasts initially at steady state in 0.4 M sorbitol survived small hypotonic shocks ($\Delta C = 0.05$ M) equally well (*Figure 2A*). However, *pil1Δ* protoplasts were more sensitive to moderate hypotonic shocks ($\Delta C = 0.1$ M) since most of them were unable to survive 2 min after the moderate shock while virtually all the WT protoplasts were able to survive (*Figure 2B*, *Figure 2—figure supplement 1*).

Moreover, eisosomes were unable to protect protoplasts from larger hypotonic shocks ($\Delta C = 0.2$ M), where most eisosomes are disassembled (*Figure 1H and I*) since most wild-type protoplasts were unable to survive longer than 4 min after these high hypotonic shocks, similarly to *pil1Δ* protoplasts (*Figure 2C*). To further confirm that increased death rate was due to the lack of eisosomes,

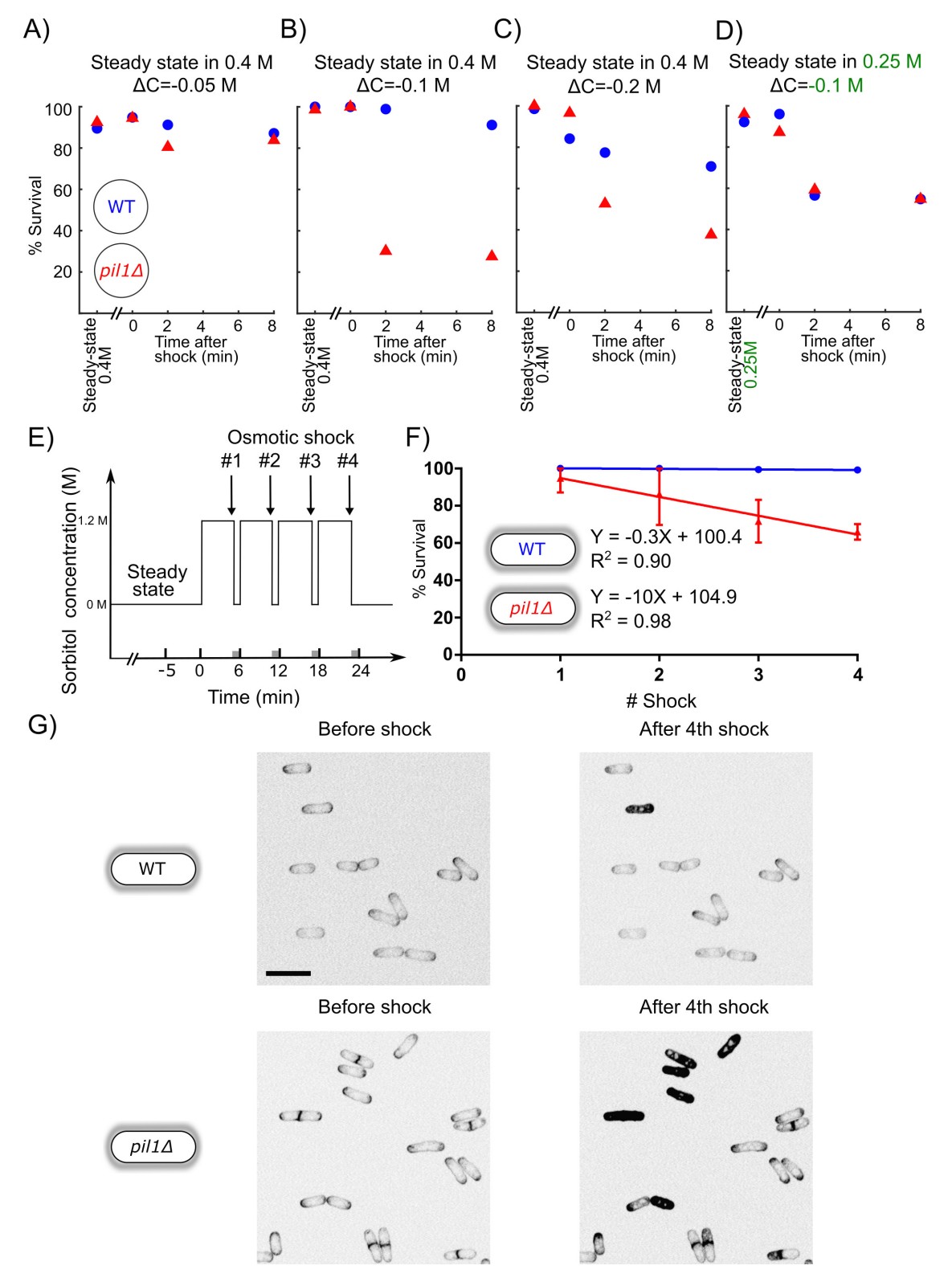

**Figure 2.** Eisosomes protect protoplasts and walled cells from osmotic shocks. (**A–C**) Percentage of wild-type (blue dots) and *pil1Δ* (red triangle) protoplasts that were alive at steady state in 0.4 M sorbitol, and after a ΔC = −0.05 M (**A**), ΔC = −0.1 M (**B**), and ΔC = −0.2 M (**C**) single hypotonic shock. Representative fields of view used to determine these percentages are shown in *Figure 2—figure supplement 1*. (**D**) Percentage of wild-type (blue dots) and *pil1Δ* (red triangles) protoplasts that were alive at steady state in 0.25 M sorbitol, and after a ΔC = −0.1 M hypotonic shock. In these

*Figure 2 continued on next page*

*Figure 2 continued*

conditions, before the shock, eisosomes in wild-type protoplasts were almost completely disassembled (*Figure 1B*). After the shock, wild-type cell survival was comparable to cells void of eisosomes because they lack Pil1p. (**E**) Timeline of repeated ΔC = 1.2 M osmotic shocks for walled cells. Each osmotic shock was performed by exchanging sorbitol concentration from 1.2 M (5 min) to 0 M (1 min). (**F**) Percentage of wild-type (blue dots, N = 273) and *pil1Δ* (red triangle, N = 197) walled cells that were alive after each osmotic shock. Note the progressive cell death induced by repeated osmotic shocks for *pil1Δ* cells. Combined data are from three independent experiments and plotted as mean ± standard deviation. (**G**) Representative images of wild-type (upper panel) and *pil1Δ* (lower panel) walled cells before shock and after the fourth shock. Dead cells were strongly stained by FM4-64 due to membrane damage. Scale bar: 10 µm.

The online version of this article includes the following source data and figure supplement(s) for figure 2:

**Source data 1.** Data for *Figure 2A*.
**Source data 2.** Data for *Figure 2B*.
**Source data 3.** Data for *Figure 2C*.
**Source data 4.** Data for *Figure 2D*.
**Source data 5.** Data for *Figure 2E*.
**Figure supplement 1.** Typical fields of view of protoplasts at steady state in 0.4 M sorbitol and 8 min after a ΔC = −0.1 M hypotonic shock.

we performed a moderate shock (ΔC = 0.1 M) on protoplasts originally at steady state in 0.25 M sorbitol, where eisosomes were mostly disassembled already. Under these conditions, WT protoplasts survival was comparable to the survival of *pil1Δ* (*Figure 2D*). This result further demonstrates that the presence of assembled eisosomes at the plasma membrane is indeed responsible for the adaptation of cells to acute hypotonic shocks, and the presence of Pil1p in the cytoplasm is not sufficient for this response.

Altogether, these experiments demonstrate that (a) eisosomes protect protoplasts from changes in their membrane tension, but only to a small extent, and (b) without eisosomes, protoplasts can withstand only minor increase in their membrane tension.

## Eisosomes protect the integrity of walled cells during consecutive osmotic shocks

We observed that a significant number of both wild-type and *pil1Δ* protoplasts died after osmotic shocks, and the percentage of *pil1Δ* protoplasts that remained alive was significantly smaller than for wild-type protoplasts even under moderate shocks of ΔC=−0.05, −0.1, and −0.2 M (*Figure 2A–C*). In contrast, we found that both wild-type and *pil1Δ* walled cells can survive a single osmotic shock of ΔC = −1.2 M, which initially led us to think that eisosomes only have a minor protective role in walled cells (*Figure 2F*). However, we noticed that subsequent osmotic shocks led to a higher mortality of *pil1Δ* compared to wild-type walled cells. While almost all the wild-type walled cells remained alive after several shocks, around 10% of *pil1Δ* walled cells died after each subsequent shock (*Figure 2E, F and G*; supplementary video 1 and 2). These results demonstrate that, even in walled cells, eisosomes exert a protective role, likely by buffering sudden changes in membrane tension.

## Membrane tension and eisosomes modulate the rate of endocytosis in cells

Within a few minutes of a hypotonic shock, the volume of WT protoplasts increased up to 50% and the volume of *pil1Δ* protoplasts increased up to 20% (*Figure 3A and B*, insets). However, the corresponding increase in surface area cannot be explained by eisosome disassembly alone – total eisosome disassembly could release about 5% of the total surface area of the plasma membrane, assuming eisosomes are hemi-cylinders with a diameter of ~50 nm and cells contain 1.6 µm of eisosomes per µm$^2$ of plasma membrane on average (*Kabeche et al., 2015*). Therefore, another mechanism for protoplasts to gain plasma membrane occurs in the first few minutes after hypotonic shocks. We hypothesized that a decrease in the number of endocytic events happening in the cell after a hypotonic shock would gradually increase the surface area of the plasma membrane and reduce membrane tension.

We measured the endocytic density, that is, the number of endocytic events in a cell normalized by the cell length, in wild-type and *pil1Δ* cells after a hypotonic shock using a ratiometric method (*Berro and Pollard, 2014a*). In brief, this method consists of imaging cells expressing a fluorescently

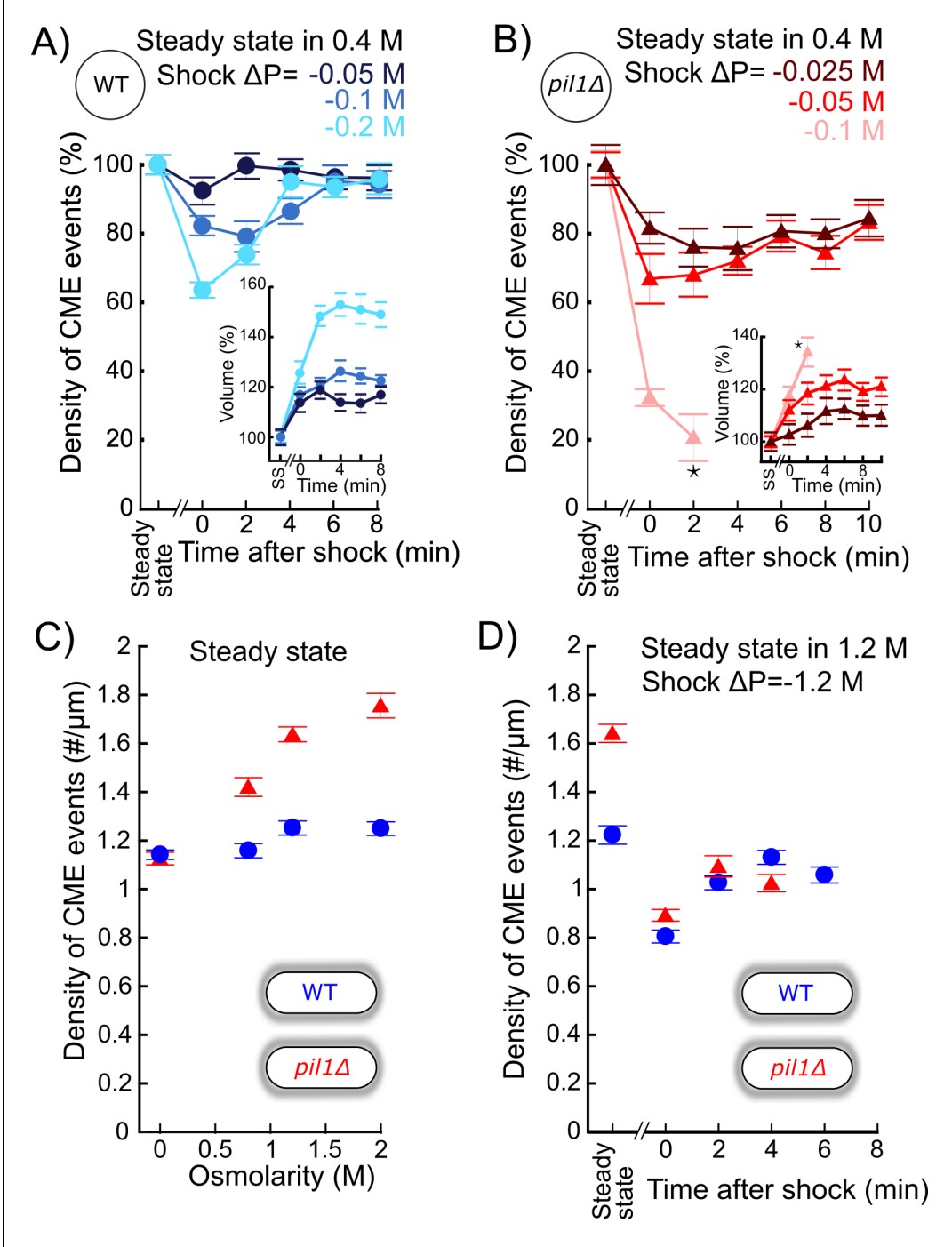

**Figure 3.** The density of endocytic events rapidly adapts after acute osmotic shocks. (**A**) Temporal evolution of density of endocytic events (average number of endocytic events at a given time in a cell divided by the cell length) in wild-type protoplasts initially at steady state in 0.4 M sorbitol and after an acute hypotonic shock of $\Delta C = -0.05$ M (dark blue, $N_{cell} \geq 102$), $\Delta C = -0.1$ M (blue, $N_{cell} \geq 54$), and $\Delta C = -0.2$ M (light blue, $N_{cell} \geq 83$). For $\Delta C = -0.1$ and $-0.2$ M, the difference in the density of clathrin-mediated endocytosis (CME) events between steady state and 0 or 2 min after the shock is statistically significant (one-way ANOVA, $p < 10^{-4}$). In all conditions, the difference after 6 min is not significant (one-way ANOVA, $p > 0.12$; details in **Figure 3—source data 1**). (**B**) Same as (**A**) but with *pil1Δ* protoplasts and hypotonic shocks of $\Delta C = -0.025$ M (dark red, $N_{cell} \geq 70$), $\Delta C = -0.05$ M (red, $N_{cell} \geq 103$), and $\Delta C = -0.1$ M (light red, $N_{cell} \geq 78$). In all conditions, the difference in the density of CME events between steady- state and any time after the shock is statistically significant (one-way ANOVA, $p < 10^{-3}$). For $\Delta C = -0.025$ and $-0.05$ M, the differences between time points after 6 min are not significant (one-way ANOVA,

*Figure 3 continued on next page*

*Figure 3 continued*

p>0.09; details in *Figure 3—source data 2*). (A and B) Insets: relative volume increase after the hypotonic shocks (the volume at steady state is used as a reference). The numbers of cells used for each condition and each time point are given in *Supplementary file 1a*. The number of cells measured in the insets is the same as that in the main figures. *: the large majority of *pil1Δ* protoplasts were too damaged or dead 4 min after the hypotonic shocks at ΔC = −0.1 M (*Figure 2B*), which prevented us to measure the density of endocytic events and the volume after this time point. (C) Density of endocytic events in intact cells at steady state in different osmolarities, $N_{cell} \geq 80$. In *pil1Δ* walled cells, the difference in the density of CME events between all pairs of conditions is statistically significant (one-way ANOVA, $p<10^{-4}$). In wild-type walled cells, the difference is small but statistically significant (details in *Figure 3—source data 3*). The numbers of cells used for each condition and each time point are given in *Supplementary file 1b*. (D) Density of endocytic events in wild-type (blue circle) and *pil1Δ* (red triangle) walled cells initially at steady state in 1.2 M sorbitol and after an acute hypotonic shock of ΔC = −1.2 M, $N_{cell} \geq 44$. The numbers of cells used for each condition and each time point are given in *Supplementary file 1c*. For wild-type and *pil1Δ* walled cells, the differences in the density of CME events after 2 min are not statistically significant (p>0.08; details in *Figure 3—source data 4*). (A), (B), (C), and (D): error bars are standard errors of the mean.

The online version of this article includes the following source data and figure supplement(s) for figure 3:

**Source data 1.** Data for *Figure 3A*.
**Source data 2.** Data for *Figure 3B*.
**Source data 3.** Data for *Figure 3C*.
**Source data 4.** Data for *Figure 3D*.
**Figure supplement 1.** Separate plots for each condition shown in *Figure 3A and B*.

tagged endocytic protein, here the actin filament crosslinking protein fimbrin (Fim1p) tagged with a monomeric enhanced green fluorescent protein (mEGFP), hereafter called Fim1p-mEGFP. First, we measure the temporal average intensity of the fluorescent protein at endocytic sites. Second, we measure the whole intensity of each cell from which the corresponding cytoplasmic intensity is subtracted – this number represents the sum of the intensities for all the fluorescently tagged proteins present at endocytic sites. The number of endocytic sites in each cell is then calculated as the ratio between those two numbers. The endocytic density is calculated by dividing this ratio with the cell length.

For all shocks tested in wild-type (ΔC = −0.05, −0.1, and −0.2 M) and *pil1Δ* protoplasts (ΔC = −0.025, −0.05, and −0.1 M) initially at steady state in 0.4 M sorbitol, the endocytic density in protoplasts significantly decreased immediately after the hypotonic shock (*Figure 3A*). The difference increased with increasing hypotonic shocks, up to 36% for wild-type protoplasts after a ΔC = −0.2 M shock, and 79% for *pil1Δ* protoplasts after a ΔC = −0.1 M shock (*Figure 3B*). These abrupt changes in the endocytic density were followed by a 2–6 min recovery back to the steady-state endocytic density, and recovery time depended on the magnitude of the hypotonic shock. Note that the change in cell volume (*Figure 3A and B*, insets) could not exclusively account for the observed decrease in the endocytic density in wild-type cells as the relative increase in cell volumes was larger than the relative decrease in endocytic density 2 min after the shocks and remained large 4 min after, while the endocytic densities recovered their pre-shock values.

Building on these results in protoplasts, we wondered whether the endocytic density in walled cells also adapts to hypotonic shocks. Indeed, immediately after the largest shock tested (ΔC = −1.2 M), we observed a similar decrease in the endocytic density for both wild-type and *pil1Δ* walled cells, 36% and 46%, respectively (*Figure 3D*). Recovery to steady-state endocytic densities occurred in less than 2 min in both wild-type and *pil1Δ* walled cells, faster than in protoplasts (*Figure 3A, B and D*). Our data show that the cell wall limits but does not completely cancel the effect of hypotonic shocks on endocytic rates. They also suggest that the regulation of the endocytic density supplements the regulation performed by the eisosomes to reduce membrane tension.

Wild-type and *pil1Δ* walled cells had a very similar adaptation after hypotonic shocks. However, we noticed a difference in the endocytic density at steady state in different sorbitol concentrations. For all concentrations tested (0–2 M), wild-type cells maintained roughly the same endocytic density. In contrast, the steady-state endocytic density in *pil1Δ* cells increased with increasing media osmolarity, up to 56% in 2 M sorbitol (*Figure 3C*). Our results suggest that eisosomes participate in

maintaining a constant density of endocytosis independently of the media osmolarity, not only after an abrupt change in membrane tension, but also when they are at steady state in different osmolarities.

## The exocytosis rate increases after a hypotonic shock in protoplasts but not in walled cells

Reciprocal to the decrease in the number of endocytic events observed after a hypotonic shock, we wondered whether the rate of exocytosis increases in the meantime to provide more surface area to the plasma membrane, as has been observed in mammalian cells (*Gauthier et al., 2009*).

To measure the rate of exocytosis in different conditions, we used the cell impermeable styryl dye FM4-64, whose fluorescence dramatically increases when it binds to membranes (*Cochilla et al., 1999*; *Gachet and Hyams, 2005*; *Richards et al., 2000*). After FM4-64 is introduced to the media, the plasma membrane is rapidly stained (*Figure 4A*). Fusion of unstained intracellular vesicles to the plasma membrane results in an increase in total cell fluorescence, because after each fusion event, a new unstained membrane from the interior of the cell is exposed to the dye. At this stage, if one wants to measure endocytosis rates, one would typically remove the FM4-64 dye from the media to destain the plasma membrane and quantify the internal fluorescence, which is proportional to the amount of membrane internalized by endocytosis. Here, we kept the FM4-64 dye in the media, and monitored the fluorescence of the plasma membrane and the interior of the cell. In this case, endo-cytic events do not increase the total cell fluorescence because they transfer patches of the plasma membrane that are already stained into the interior of the cell (*Figure 4A*, red arrow). Therefore, the increase in fluorescence we measured is due to the addition of new unstained lipids to the plasma membrane by exocytosis (*Figure 4A*, gray arrow). Note that the increase in total cell fluorescence could also be due to putative transfer of lipids by non-exocytic mechanisms (*Reinisch and Prinz, 2021*), but for simplicity and by lack of further evidence, hereafter, we will interpret the increase in fluorescence to an increase in exocytosis rate.

Staining of wild-type fission yeast with 20 µM FM4-64 in EMM5S (*Figure 4B*) showed that after a brief phase of rapid staining of the cell surface, the total cell fluorescence intensity grows roughly lin-early for at least 20 min, and the slope of the normalized intensity corresponds to the exocytosis rate as a percentage of the plasma membrane surface area per unit of time (see 'Materials and methods') (*Gauthier et al., 2009*; *Smith and Betz, 1996*; *Vida and Emr, 1995*). Using this method, we measured that wild-type walled cells at steady state in EMM5S exocytose 4.6% of their plasma membrane surface area per minute (*Figure 4B*). FM4-64 staining did not seem to affect the endo-cytic and exocytic membrane trafficking of yeast cells, since stained vesicles were successfully released after washing cells with fresh media (*Figure 4B*).

We measured the exocytosis rates in the conditions that had the largest effects on the rates of endocytosis while keeping most cells alive, i.e., we used protoplasts at steady state in 0.4 M and per-formed a $\Delta C = -0.2$ M shock for wild-type and $\Delta C = -0.05$ M shock for *pil1Δ*. At steady state in 0.4 M sorbitol (*Figure 4F and H*), wild-type protoplasts had an exocytosis rate similar to walled cells in EMM5S in 0 M sorbitol ($k_{0-5}=4.4 \pm 0.2\%$ min$^{-1}$). After a $\Delta C = -0.2$ M shock, the exocytosis rate increased by 41% ($k_{0-5}=6.2 \pm 0.4\%$ min$^{-1}$). At steady state in 0.4 M sorbitol (*Figure 4G and H*), the exocytosis rate of *pil1Δ* protoplasts was higher than that for walled cells in 0 M sorbitol ($k_{0-5}=6.2 \pm 0.4\%$ min$^{-1}$). After a $\Delta C = -0.05$ M shock, the exocytosis rate increased modestly ($k_{0-5}=6.8 \pm 0.5\%$ min$^{-1}$). Therefore, in both wild-type and *pil1Δ* protoplasts, an acute hypotonic shock leads to an increased exocytosis rate, which increases surface area and likely reduces membrane tension. The more modest change in exocytosis rate in *pil1Δ* protoplasts than in wild-type cells highlights the role of eisosomes in buffering the change in the exocytosis rate in response to changes in osmolarity and membrane tension.

We wondered whether these changes in exocytosis rate also happen in walled cells. First, we measured exocytosis rate at steady state in solutions with different molarities and found that the rates were smaller than in protoplasts (*Figure 4C–E*). The exocytosis rate of wild-type walled cells at steady state in 1.2 M sorbitol ($k_{0-5}=3.1 \pm 0.1 \%$ min$^{-1}$, *Figure 4C and E*) was 35% smaller than in 0 M sorbitol ($k_{0-5}=4.8 \pm 0.1 \%$ min$^{-1}$, *Figure 4B*). In addition, in *pil1Δ* walled cells, the exocytosis rate of walled cells lacking eisosomes in 1.2 M sorbitol was only slightly smaller than wild-type cells in the same conditions ($k_{0-5}=2.6 \pm 0.1\%$ min$^{-1}$, *Figure 4D and E*). After hypotonic shocks, the change in exocytosis rate in walled cells was very limited (*Figure 4C–E*). In fact, our strongest hypotonic shock

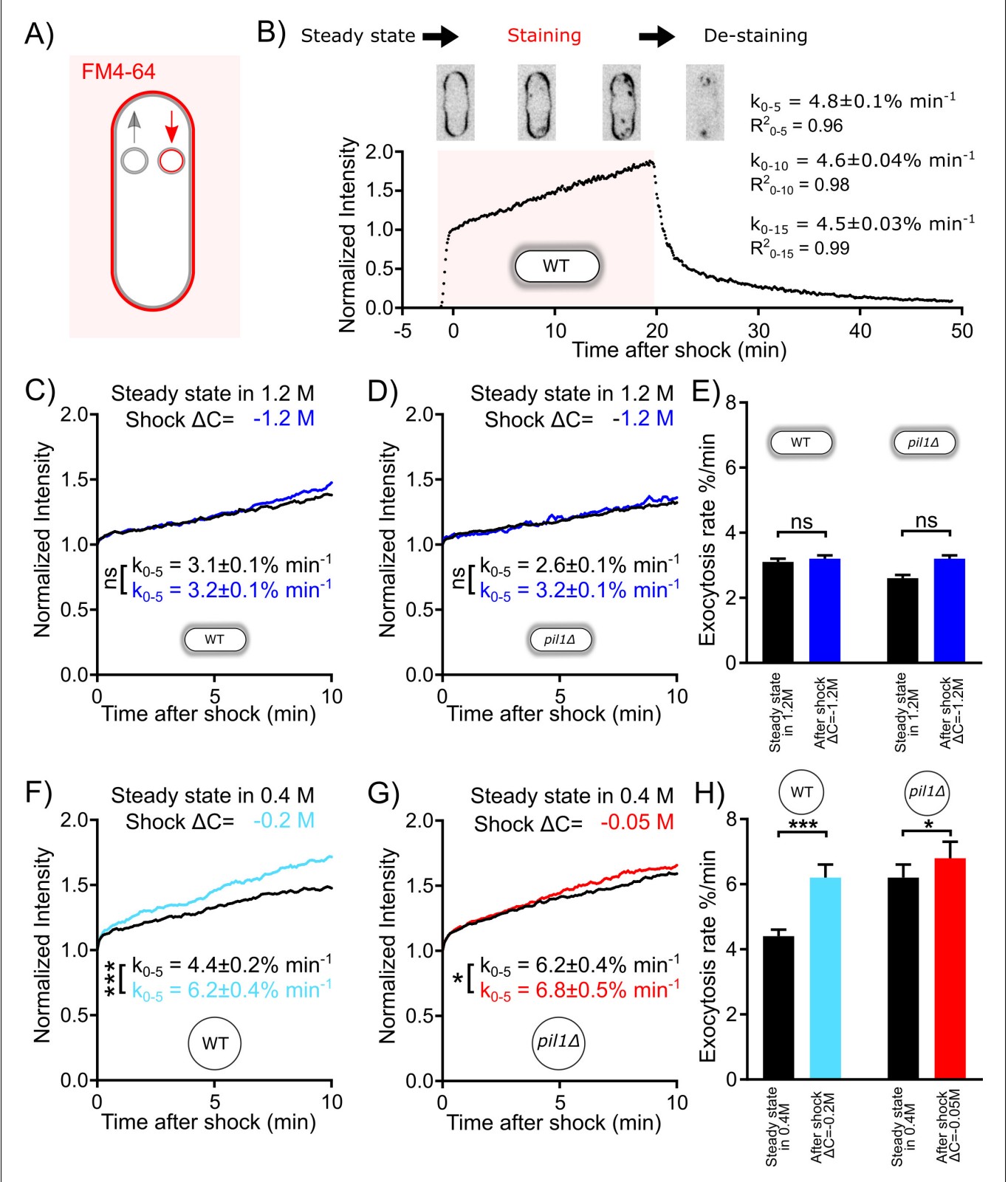

**Figure 4.** Exocytosis rate increases after an acute change in membrane tension in protoplasts but not in walled cells. (**A**) Rationale of measurement of whole cell exocytosis rate through FM4-64 staining. After FM4-64 is flown in the imaging chamber, the dye rapidly binds to the cell surface in less than a minute. After this initial phase, the whole cell fluorescence increases every time the new (unlabeled) internal membrane is exposed to the cell surface by exocytosis. Note that endocytic events do not change the total fluorescence measured. (**B**) Measurement of yeast cell exocytosis rate at steady state

*Figure 4 continued on next page*

*Figure 4 continued*

in 0 M sorbitol. Cells were stained with 20 μM FM4-64 in Edinburgh Minimum Media (EMM5S) for 20 min before washing with EMM5S. During FM4-64 staining, the fluorescence intensity increases rapidly for 1 min before entering a slow linear phase over at least 20 min for wild-type cells. The fluorescence intensity at the end of the initial rapid increase phase corresponds to the complete staining of cell surface. It was normalized to 1, so that the subsequent increase in fluorescence intensity corresponds to a percentage of the plasma membrane surface area. After the dye was removed 20 min later, the decrease in fluorescence intensity suggests that the incorporation of FM4-64 didn't interfere with the vesicle trafficking pathway of the cell. The rate of exocytosis (measured as a percentage of the plasma membrane surface area per minute) is the slope of a linear fit of the measured signal over the first 5 min ($k_{0-5}$), 10 min ($k_{0-10}$), or 15 min ($k_{0-15}$). Example images of stained cells at different time points are shown in the middle panel (inverted contrast). (C–H) Rates of exocytosis at steady state and after hypotonic shocks. (C and D) The exocytic rate of wild-type walled cells is not changed after a $\Delta C = -1.2$ M acute hypotonic shock (black, before shock, $N_{cells} = 79$; blue, after shock, $N_{cells} = 68$; three replicates each). The exocytic rate of *pil1Δ* walled cells does not change significantly in the same conditions (black, before shock, $N_{cells} = 60$; blue, after shock, $N_{cells} = 96$; three replicates each). All walled cells were at steady state in 1.2 M sorbitol before time 0 min. Curves for individual conditions in panels (C) and (D) are plotted in *Figure 4—figure supplements 1A,B*, respectively. (E) Summary of exocytic rates for wild-type and *pil1Δ* walled cells before and after hypotonic shock. (F and G) The exocytic rate of wild-type and *pil1Δ* protoplasts increases after $\Delta C = -0.2$ M (black, before shock, $N_{cells} = 20$; light blue, after shock, $N_{cells} = 37$; four replicates each) and $\Delta C = -0.05$ M (black, before shock, $N_{cells} = 44$; red, after shock, $N_{cells} = 60$; four replicates each) acute hypotonic shocks, respectively. Before time 0 min, all protoplasts were at steady state in 0.4 M sorbitol. Curves for individual conditions in panels (F) and (G) are plotted in *Figure 4—figure supplements 1C,D*, respectively. (H) Summary of exocytic rates for wild-type and *pil1Δ* protoplasts before and after hypotonic shock. (C–H) Data from at least three independent experiments were pooled together to produce each curve. p-values: non-significant (ns), p>0.05; *p≤0.05; ***p≤0.001.

The online version of this article includes the following source data and figure supplement(s) for figure 4:

**Source data 1.** Data for *Figure 4B*.
**Source data 2.** Data for *Figure 4C*.
**Source data 3.** Data for *Figure 4D*.
**Source data 4.** Data for *Figure 4E*.
**Source data 5.** Data for *Figure 4F*.
**Source data 6.** Data for *Figure 4G*.
**Source data 7.** Data for *Figure 4H*.
**Figure supplement 1.** Separate plots for each condition in *Figure 4*.

of $\Delta C = -1.2$ M did not significantly increase the exocytosis rate of wild-type or *pil1Δ* walled cells (*Figure 4E*). These data corroborate our previous finding that the cell wall limits but does not completely cancel the effect of hypotonic shocks in intact cells. In addition, they also demonstrate that eisosomes are involved in the regulation of the exocytosis rate.

## Inhibition of exocytosis decreased the survival rate of protoplasts under acute hypotonic shocks and inhibition of endocytosis increased their survival rate

To further test our hypothesis that reducing the endocytosis rate and increasing the exocytosis rate help regulate membrane tension after a hypotonic shock, we blocked endocytosis or exocytosis with drugs and measured the survival rates of cells. We hypothesized that inhibition of endocytosis or exocytosis would have opposite effects on the survival of protoplasts under acute hypotonic shocks. Specifically, inhibition of endocytosis would help retain membranes on the surface of protoplasts, thereby reducing the probability of membrane rupture. Conversely, inhibition of exocytosis would reduce the transfer of membranes from intracellular vesicles to the surface of protoplasts, exacerbating the lack of plasma membrane in the face of imminent protoplast expansion. To observe the largest effects, we used *pil1Δ* protoplasts under $\Delta C = -0.2$ M shock and exposed the cells to either Latrunculin A (LatA) or Brefeldin A (BFA) for 30 min before the shocks.

Blocking exocytosis with BFA increased the death rate of protoplasts after hypotonic shocks, confirming our hypothesis (*Figure 5*). Blocking actin assembly, and therefore endocytosis, with LatA made the protoplasts more resistant starting 4 min after the hypotonic shock, also confirming our hypothesis. Note that LatA treatment made the protoplasts less resistant to shock in the initial 2 min after the hypotonic shock, which seems in contradiction with our hypothesis. However, it is possible that prolonged treatment with LatA had other unidentified effects on protoplast survival or may indirectly affect the exocytosis rate since LatA affects all actin structures in the cell, including actin cables, which are needed for the transport of exocytic vesicles (*Lo Presti et al., 2012*).

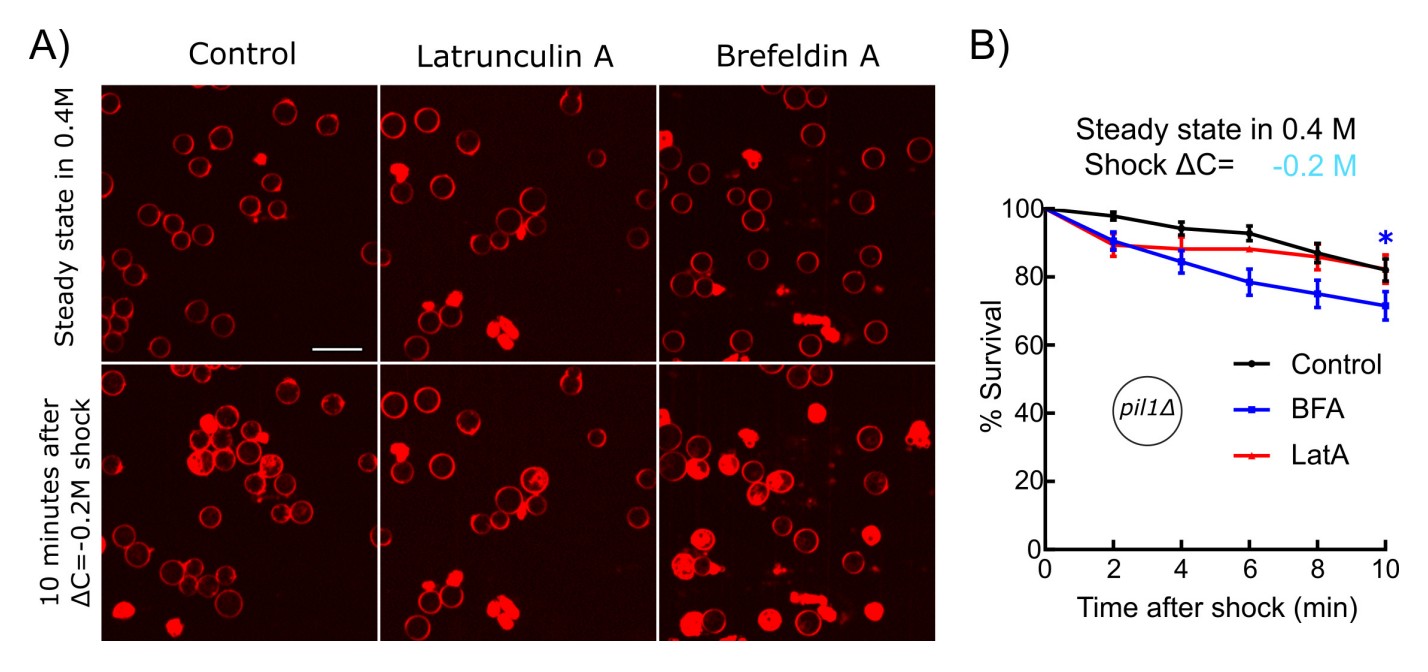

**Figure 5.** Inhibition of exocytosis but not endocytosis decreased the survival rate of protoplasts under acute hypotonic shocks. (A) *pil1Δ* protoplasts initially at steady state in 0.4 M sorbitol and supplemented with 25 µM Latrunculin A (LatA) (middle column) or 2 mM Brefeldin A (BFA) (right column) or nothing (control, left column) were submitted to a ΔC = −0.2 M hypotonic shock (t = 0 min). Cells were considered dead if they contained large amounts of intracellular red fluorescence from the FM4-64 dye, which is the consequence of a rupture of the plasma membrane. Scale bar: 10 µm. (B) Survival rate for all conditions. Black: control (N = 114); blue: 2 mM BFA (N = 83); red: 25 µM Latrunculin A (N = 70). Data were pooled from two independent experiments and plotted as Kaplan-Meier survival curves. Error bars: standard error of the mean by the Greenwood formula. *p≤0.05, logrank test.

The online version of this article includes the following source data for figure 5:

**Source data 1.** Data for *Figure 5B*.

## Actin dynamics during clathrin-mediated endocytosis in wild-type walled cells is robust over a wide range of chronic and acute changes in media osmolarity

Next, we wondered how the actin machinery adapts to different changes in osmotic pressure and membrane tension. To monitor actin dynamics during CME, we imaged fission yeast cells expressing Fim1p-mEGFP (*Figure 6A and B*). Fimbrin is a bona fide marker for endocytosis in yeast since it has spatial and temporal co-localization with the classical endocytic marker End4p (the fission yeast homolog of mammalian Hip1R and budding yeast Sla2) during endocytosis (*Figure 6—figure supplements 1A,B*). Fimbrin's time of appearance, disappearance, peak number of molecules, and spatial localization follow those of actin in wild-type cells and all mutants tested so far (*Arasada et al., 2018*; *Berro and Pollard, 2014b*; *Chen and Pollard, 2013*; *Sirotkin et al., 2010*). Fimbrin is the most abundant endocytic protein that is fully functional when tagged with a fluorescent protein at either N- or C-terminal. Tagged fimbrin is a more robust marker for actin dynamics than tagged actin or actin-binding markers such as LifeAct or calponin-homology domains, because they require overexpression, which is difficult to control precisely in fission yeast and potentially creates artifacts (*Courtemanche et al., 2016*; *Suarez et al., 2015*). Fimbrin is also a central player in force production during CME in yeast (*Ma and Berro, 2019*; *Ma and Berro, 2018*; *Picco et al., 2018*; *Planade et al., 2019*). We optimized our imaging protocols, and improved tracking tools and temporal super-resolution alignment methods (*Berro and Pollard, 2014a*) to (a) easily collect hundreds of endocytic events in an unbiased manner and (b) achieve high reproducibility between different samples, fields of view, and days of experiment (*Figure 6C*, *Figure 6—figure supplements 2A*). These improvements in our quantitative microscopy protocol have allowed us to detect small

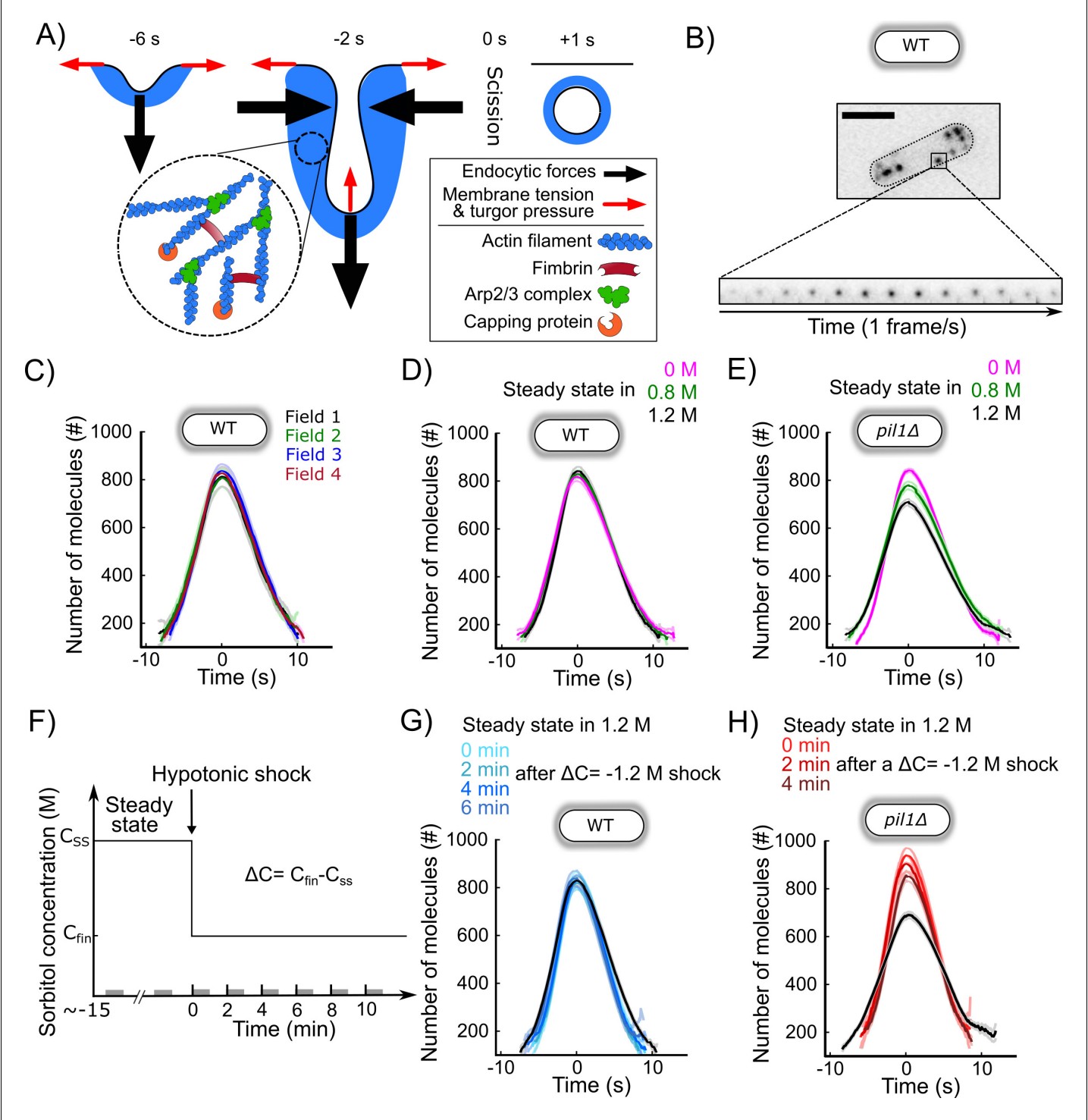

**Figure 6.** CME in walled cells is robust over a wide range of conditions. (**A**) Schematic of the plasma membrane deformations and the main components of the actin machinery during clathrin-mediated endocytosis (CME). Fimbrin (Fim1p, red) crosslinks actin filaments (blue) present at endocytic sites and is used as a proxy to monitor the amount of actin assembled (***Figure 6—figure supplement 1***). (**B**) Wild-type walled fission yeast cell expressing Fim1p-mEGFP (inverted contrast). Top: cell outlined with a dashed line; scale bar: 5 μm. Bottom: montage of a representative CME event. The interval between each frame is 1 s. (**C**) The number of molecules of Fim1p-mEGFP detected, tracked, and aligned with temporal super-resolution (***Berro and Pollard, 2014a***) is highly reproducible between fields of view (one-way ANOVA on the number of molecules at time 0 s, p=0.74). Each curve with a dark color represents the average of several endocytic events from a different field of view of the same sample (N ≥ 64), and the light colors are the 95% confidence intervals. For each average curve, the peak value corresponds to time 0 s, when vesicle scission happens. (**D**) Number of

*Figure 6 continued on next page*

*Figure 6 continued*

molecules of Fim1p-mEGFP in wild-type walled cells at steady state in media supplemented with different sorbitol concentrations. There is no statistically significant difference in the number of molecules at time 0 s between the three conditions (one-way ANOVA, p=0.29). N ≥ 388. (E) Number of molecules of Fim1p-mEGFP in *pil1Δ* walled cells at steady state in media supplemented with different sorbitol concentrations (N ≥ 342). The difference in the number of molecules at time 0 s between all pairs of conditions is statistically significant (one-way ANOVA, $p<10^{-5}$). (F) Timeline of the hypotonic shock experiments and notations. By convention, hypotonic shocks start at time 0 min and are defined by the difference in concentrations of sorbitol in the steady-state media before the shock ($C_{SS}$) and after the hypotonic shock ($C_{fin}$), $\Delta C = C_{fin} - C_{SS}$. Data for a given time point correspond to endocytic events happening within 1 min after this time point (e.g., the data at t = 0 min correspond to endocytic events happening between 0 and 1 min after the shock). These time intervals are represented by gray bars on the time axis. (G) Number of molecules of Fim1p-mEGFP for wild-type walled cells initially at steady state in 1.2 M sorbitol and after an acute osmotic shock of $\Delta C = -1.2$ M. There is no statistically significant difference in the number of molecules at time 0 s between the three conditions (one-way ANOVA, p=0.95). Black: steady state in 1.2 M sorbitol; light to dark blue in top panel: 0, 2, 4, and 6 min after the acute hypotonic shock (N ≥ 103). (H) Number of molecules of Fim1p-mEGFP in *pil1Δ* walled cells before and after an acute osmotic shock ($\Delta C = -1.2$ M). The difference in the number of molecules at time 0 s between all pairs of conditions is statistically significant (one-way ANOVA, p<0.03) except between 0 and 2 min after the shock (one-way ANOVA, p=0.18). Black: steady state in 1.2 M sorbitol before the hypotonic shock (N = 583); light to dark red in top panel: 0, 2, and 4 min after the acute hypotonic shock (N ≥ 145). (C, D, E, G, and H) Dark colors: average; light colors: average ± 95% confidence interval. The number of molecules and speed versus time for each condition are plotted separately in *Figure 6— figure supplements 2C*, *Figure 6—figure supplements 3D,G*, and *Figure 6—figure supplement 4E,H* (E and H). The numbers of endocytic events used in each curve are given in *Supplementary file 1d* (C), *Supplementary file 1e* (D), *Supplementary file 1f* (E), *Supplementary file 1g* (G), and *Supplementary file 1h* (H).

The online version of this article includes the following source data and figure supplement(s) for figure 6:

**Source data 1.** Data for *Figure 6C* and for *Figure 6—figure supplement 2*.
**Source data 2.** Data for *Figure 6D* and for *Figure 6—figure supplement 3A*.
**Source data 3.** Data for *Figure 6E* and for *Figure 6—figure supplement 4A*.
**Source data 4.** Data for *Figure 6G* and for *Figure 6—figure supplement 3B*.
**Source data 5.** Data for *Figure 6H* and for *Figure 6—figure supplement 4B*.
**Figure supplement 1.** Fimbrin is a proxy for actin dynamics during CME in yeast, and eisosomes do not participate in CME in wild-type walled cells.
**Figure supplement 2.** Speed data (A) and separate plots (B) for the data from each field of view in panel (1C).
**Figure supplement 3.** Representative endocytic events, speeds, and separate plots for each condition in *Figure 6D and G*.
**Figure supplement 4.** Representative endocytic events, speeds, and separate plots for each condition in *Figure 6E and H*.

differences between mutants or conditions that would have been missed with previous methods. We confirmed that Fim1p accumulates at endocytic sites for about 10 s and then disassembles while the vesicle diffuses away from the plasma membrane (*Figure 6C*, *Figure 6—figure supplements 2A*; *Sirotkin et al., 2010*; *Skau et al., 2011*). As a convention, the peak number of Fim1p molecules is set to time 0 s and corresponds to vesicle scission in intact wild-type cells (*Berro and Pollard, 2014a*; *Berro and Pollard, 2014b*; *Sirotkin et al., 2010*).

For all tested osmolarities at steady state in walled wild-type cells, we observed no significant difference in the dynamics of fimbrin recruitment or disassembly, maximum molecule number, or endocytic patch movements (*Figure 6D*). Our results indicate that wild-type walled cells have adaptation mechanisms for chronic exposure to a wide range of osmolarities, which allow them to perform CME in a highly reproducible manner.

We then tested the robustness of the endocytic actin machinery when cells experienced a hypotonic shock, which aimed to abruptly increase the tension of their plasma membrane. To observe the highest possible effect, we imaged cells grown at steady state in 1.2 M sorbitol and rapidly exchanged the media with a buffer free of sorbitol (*Figure 6F and G*), therefore performing an acute hypotonic shock of $\Delta C = -1.2$ M. Despite the high hypotonic shock, which represents a ~3 MPa drop in pressure, CME proceeded quite similarly to steady-state conditions (*Figure 6G*, *Figure 6— figure supplements 3*). The maximum number of fimbrin proteins was the same before and after the hypotonic shock, but fimbrin assembly and disassembly were ~15% faster after the shock.

## Eisosomes mitigate the response of the endocytic machinery to acute and chronic changes in media osmolarity

The robustness of the endocytic process under a wide range of chronic and acute exposure to different media osmolarities is consistent with our previous results showing that fission yeast cells rapidly regulate plasma membrane tension. To better understand the role of eisosomes and amplify the

change in membrane tension after hypotonic shocks, we repeated our experiments in *pil1Δ* cells that lack eisosomes (*Figure 6E*).

Dynamics of Fim1p during CME for wild-type and *pil1Δ* walled cells at steady state in media free of sorbitol was identical (*Figure 6—figure supplement 1C*). However, at steady state in media with high sorbitol concentrations, cells lacking eisosomes recruited slightly fewer fimbrin molecules to endocytic patches than wild-type cells (*Figure 6E*). The maximum number of Fim1p assembled at CME sites in *pil1Δ* cells in buffers containing 0.8 and 1.2 M sorbitol was 10 and 17% lower, respectively. Within the first 2 min of an acute hypotonic shock from 1.2 M sorbitol to 0 M ($\Delta C = -1.2$ M), the maximum number of Fim1p increased by 30%, while its timing was shortened by ~30% compared to steady state (*Figure 6H*). Four minutes after the hypotonic shock, the dynamics of fimbrin stabilized at its steady-state dynamics in 0 M sorbitol (*Figure 6E and H*). Overall, our data show that the endocytic actin machinery in cells lacking eisosomes is more sensitive to acute and chronic changes in media osmolarity than in wild-type cells, consistent with a role for eisosome in regulating membrane tension at endocytic sites.

## CME in protoplasts is sensitive to chronic changes in osmolarity

Endocytosis in wild-type protoplasts at steady state in a medium containing 0.4 or 0.8 M sorbitol was able to proceed normally by recruiting almost the same number of fimbrin molecules as in walled cells, but with a slightly longer timing (*Figure 7—figure supplement 1B*). In contrast, in a medium with 1.2 M sorbitol, the timing of fimbrin recruitment was dramatically longer, and endocytosis failed to proceed normally, as reported by the virtually null speed of patches during the entire time fimbrin was present at the endocytic site (*Figure 7—figure supplement 1B*). Cells lacking eisosomes had very similar phenotypes but endocytosis started failing at 0.8 M sorbitol (*Figure 7—figure supplement 1C*).

At 0.25 M sorbitol, both wild-type and *pil1Δ* protoplasts were able to perform endocytosis but required a larger amount of Fim1p (*Figure 7—figure supplements 1B,C*). In these conditions, the eisosomes covered only half of the plasma membrane surface area they cover at 0.4 M sorbitol (*Figure 1B and C*), and our data suggest that the plasma membrane was under high tension (*Figure 1G*). This result indicates that during CME, the actin machinery is able to adapt to mechanical cues by mechanisms that are independent of the cell wall.

For both wild-type and *pil1Δ* protoplasts in 0.4 M sorbitol, the temporal evolution of the number of fimbrin molecules and the speed of patches were close to the same metrics measured in walled cells in media without sorbitol (*Figure 7—figure supplements 1B,C*). These results suggest that the osmotic pressure at these concentrations, which is equivalent to a pressure of 1 MPa, is close to the naturally maintained turgor pressure of walled fission yeast cells, in good agreement with previous measurements (*Minc et al., 2009*). Therefore, to keep protoplasts in conditions close to walled cells, the steady-state media used in our following experiments on protoplasts contained 0.4 M sorbitol.

## The endocytic actin machinery rapidly adapts to increases in membrane tension

To characterize the adaptation of the endocytic actin machinery to a rapid increase in turgor pressure and membrane tension, we repeated our acute hypotonic shocks ($\Delta C = -0.05$, $-0.1$, or $-0.2$ M) on protoplasts initially at steady state in media containing 0.4 M sorbitol. After low ($\Delta C = -0.05$ M) and medium ($\Delta C = -0.1$ M) acute shocks in wild-type protoplasts, we did not observe any stalled endocytic events – when cells started the recruitment of the actin machinery, endocytosis proceeded to successful completion (*Figure 7A–C* and *Figure 7—figure supplements 5*). The recruitment of fimbrin did not significantly change over time. In contrast, 2 min after a $\Delta C = -0.2$ M shock, endocytic sites recruited 20% more fimbrin, and it took ~25% longer to perform endocytosis (*Figures 7C, D* and *Figure 7—figure supplements 5*). The actin machinery restored its steady-state behavior less than 4 min after the shock (*Figure 7D*).

We repeated these experiments with *pil1Δ* protoplasts to eliminate the role of eisosomes in the reduction of membrane tension during hypotonic shocks. Immediately (0 min) after the lowest hypotonic shock tested ($\Delta C = -0.05$ M), fimbrin recruitment took slightly longer and the number of proteins recruited was higher than at steady state (*Figure 7E–G* and *Figure 7—figure supplements 6*). While fimbrin restored its steady-state dynamics in less than 4 min after a high acute hypotonic shock

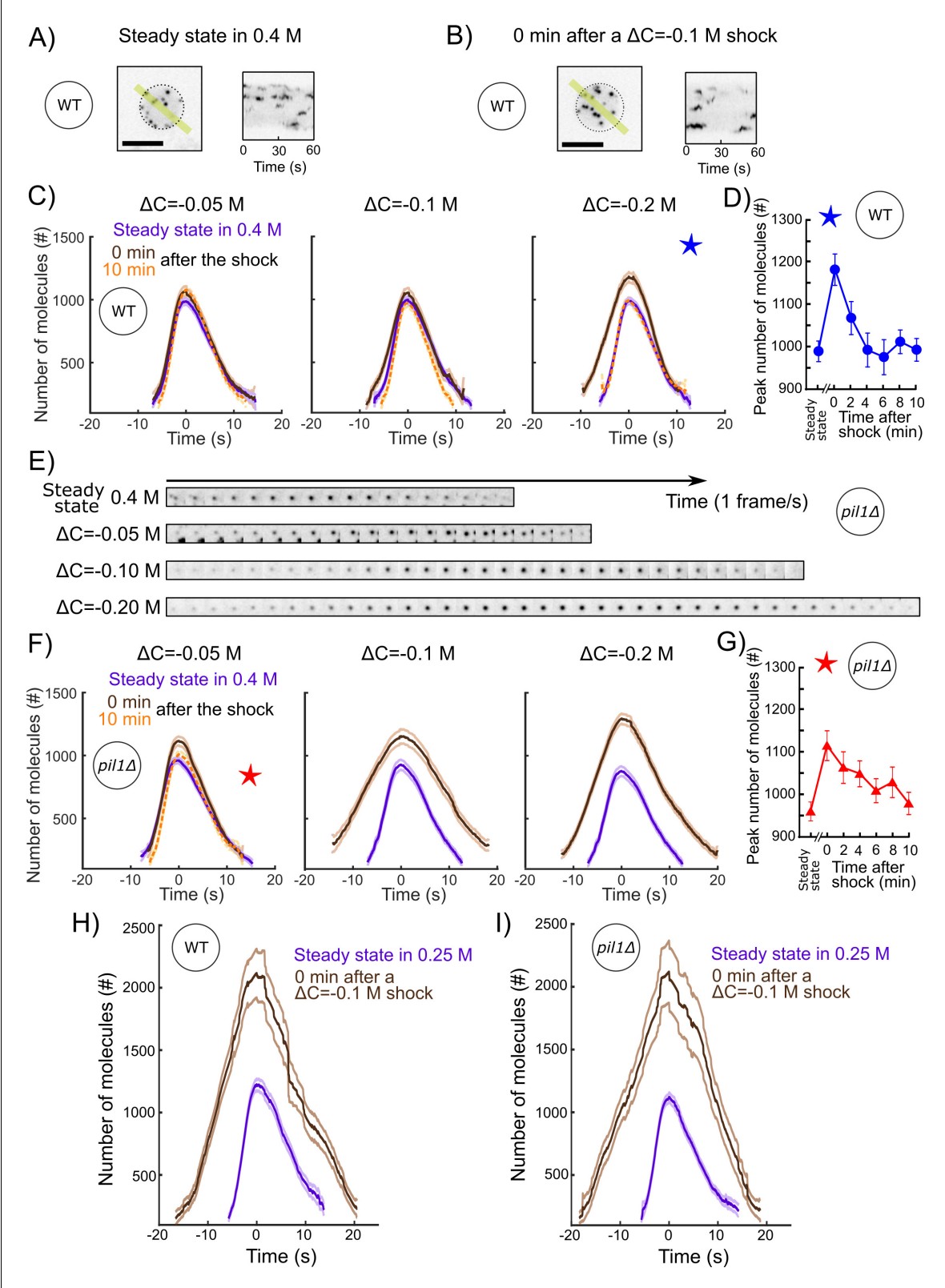

**Figure 7.** The actin endocytic machinery adapts to increases of membrane tension in protoplasts. (**A and B**) Representative wild-type protoplasts expressing Fim1-mEGFP (inverted contrast) at steady state in 0.25 M sorbitol (A, left panel) and immediately after (0 min) an acute osmotic shock of ΔC = −0.1 M (B, left panel). Right panels: kymographs of the fluorescence under the yellow lines in the left panels. Black dashed lines: protoplast outline. Scale bars: 5 μm. (**C and F**) Number of Fim1p-mEGFP molecules in wild-type (**C**) and *pil1Δ* (**F**) protoplasts at steady state in 0.4 M sorbitol

*Figure 7 continued on next page*

*Figure 7 continued*

(purple), 0 min (brown) and 10 min (orange) after a hypotonic shock of $\Delta C = -0.05$ M (left panels), $\Delta C = -0.1$ M (middle panels), and $\Delta C = -0.2$ M (right panels), N $\geq$ 95. Data for each condition are plotted separately in *Figure 7—figure supplement 3* (wild type) and *Figure 7—figure supplement 4* (*pil1Δ*). The speeds of Fim1p-mEGFP for each condition are shown in *Figure 7—figure supplement 5* (wild type) and *Figure 7—figure supplement 6* (*pil1Δ*). The numbers of endocytic events used in each curve are given in *Supplementary file 1i*. Note that the large majority of *pil1Δ* protoplasts were too damaged or dead 2 min after hypotonic shocks larger than or equal to $\Delta C = -0.1$ M to allow us to track enough endocytic events and produce a curve (*Figure 2B,C* and *Figure 7—figure supplement 1*). In panel (C), the difference in the number of molecules at time 0 s at steady state and 0 min after the shock is statistically significant for all shocks (one-way ANOVA, p<0.03), and the difference between steady state and 10 min after the shock is not statistically significant (one-way ANOVA, p>0.2; details in the data file for C). In panel (F), the difference at steady state and 0 min after the shock is statistically significant for all shocks (one-way ANOVA, p<$10^{-5}$; details in the data file for *Figure 5F*). (D) Temporal adaptation of the peak number of Fim1p-mEGFP in wild-type protoplasts initially at steady state in 0.4 M sorbitol and 0–10 min after a $\Delta C = -0.2$ M osmotic shock. The condition for this figure is the same as the condition with the blue star in (C). The difference between steady state and 0 or 2 min after shock is statistically significant (one-way ANOVA, p<$10^{-3}$; details in the data file for D). The difference between steady state and 4, 6, 8, and 10 min after shock is not statistically significant (one-way ANOVA, p>0.2; details in the data file for D). (E) Montage of representative endocytic events (Fim1-mEGFP, inverted contrast) in *pil1Δ* protoplasts (one frame per second) at steady state in 0.4 M sorbitol (first row) and immediately after (0 min) hypotonic shocks of $\Delta C = -0.05$ M (second row), $\Delta C = -0.10$ M (third row), and $\Delta C = -0.20$ M (fourth row). (G) Temporal adaptation of the peak number of Fim1p-mEGFP in *pil1Δ* protoplasts initially at steady state in 0.4 M sorbitol and 0–10 min after a $\Delta C = -0.05$ M shock. The condition in this figure is the same as the condition with the red star in (F). The difference between steady state and 0, 2, 4, 6, or 8 min after shock is statistically significant (one-way ANOVA, p<0.01; details in the data file for F). The difference between steady state and 10 min after shock is not statistically significant (one-way ANOVA, p>0.3; details in the data file for F). (D and G) Error bars are 95% confidence intervals. The numbers of endocytic events at each time point are given in *Supplementary file 1j*. (H and I) Number of molecules of Fim1p-mEGFP for wild-type (H) and *pil1Δ* (I) protoplasts at steady state in 0.25 M sorbitol (purple dashed) and immediately after (0 min) a hypotonic shock of $\Delta C = -0.1$ M (brown), N $\geq$ 67. The difference in the number of molecules at time 0 s at steady state and 0 min after the shock is statistically significant for all conditions (one-way ANOVA, p<$10^{-16}$). The speed data for each condition are plotted in *Figure 7—figure supplement 7*. The numbers of endocytic events used in each curve are given in *Supplementary file 1k*. The survival rates for the wild-type and *pil1Δ* protoplasts in these conditions are plotted in *Figure 2D*.

The online version of this article includes the following source data and figure supplement(s) for figure 7:

**Source data 1.** Data for *Figure 7C*.
**Source data 2.** Data for *Figure 7D*.
**Source data 3.** Data for *Figure 7F*.
**Source data 4.** Data for *Figure 7G*.
**Source data 5.** Data for *Figure 7H*.
**Source data 6.** Data for *Figure 7I*.
**Figure supplement 1.** CME in protoplasts at steady state in different sorbitol concentrations.
**Figure supplement 1—source data 1.** Data for *Figure 7—figure supplement 1B*.
**Figure supplement 1—source data 2.** Data for *Figure 7—figure supplement 1C*.
**Figure supplement 2.** Separate plots for each condition in *Figure 7—figure supplement 1*.
**Figure supplement 3.** Separate plots for each condition shown in *Figure 7C*.
**Figure supplement 4.** Separate plots for each condition shown in *Figure 7F*.
**Figure supplement 5.** Speeds and separate plots for each condition shown in *Figure 7C*.
**Figure supplement 5—source data 1.** Data for *Figure 7—figure supplement 5*.
**Figure supplement 6.** Speeds and separate plots for each condition shown in *Figure 7F*.
**Figure supplement 6—source data 1.** Data for *Figure 7—figure supplement 6*.
**Figure supplement 7.** Speed of Fim1p-mEGFP at CME sites for wild-type (A) and *pil1Δ* (B) protoplasts at steady-state in 0.25 M sorbitol (purple) and immediately (0 min) after (brown) a hypotonic shock of $\Delta C = -0.1$ M.
**Figure supplement 7—source data 1.** Data for *Figure 7—figure supplement 7*.

($\Delta C = -0.2$ M) in wild-type protoplasts (*Figure 7D*), recovery of fimbrin dynamics to its steady-state behavior in *pil1Δ* protoplasts occurred over 10 min, even for the most modest hypotonic shock, $\Delta C = -0.05$ M (*Figure 7G*). The changes in fimbrin dynamics in *pil1Δ* protoplasts became increasingly larger for $\Delta C = -0.1$ and $-0.2$ M hypotonic shocks – endocytic sites assembled a peak number of fimbrin, respectively, ~25 and ~50% larger and took ~85 and ~50% longer.

Wild-type protoplasts at steady state in 0.25 M sorbitol contained significantly fewer assembled eisosomes despite expressing normal amounts of Pil1p (*Figure 1B, C and E*). We took advantage of this condition to test whether the absence of eisosome structures at the plasma membrane, and not the absence of the protein Pil1p, is responsible for changes in actin dynamics after an acute hypotonic shock. We subjected wild-type protoplasts at steady state in 0.25 M sorbitol to an acute hypotonic shock of $\Delta C = -0.1$ M (*Figure 7H and I*). Two minutes after the shock, endocytic sites

accumulated 73% more fimbrin and took ~60% longer (*Figure 7H* and *Figure 7—figure supplement 7A*). This behavior was nearly identical to fimbrin dynamics in *pil1Δ* protoplasts under the same conditions (*Figure 7I* and *Figure 7—figure supplement 7B*).

Altogether, these experiments demonstrate that (a) the endocytic actin machinery adapts to compensate the increase in membrane tension and (b) actin dynamics restores its steady-state behavior within a few minutes, providing the protoplasts survived the hypotonic shock.

## Discussion

### Mechanisms of tension regulation and homeostasis of the plasma membrane

Our results demonstrate that the regulation of membrane tension in hypotonic environment is performed via a combination of at least three mechanisms: the mechanical protection by the cell wall, the disassembly of eisosomes, and the temporary shift in the balance between endocytosis and exocytosis (*Figure 8*). Our data indicate that all three mechanisms are used in parallel since wild-type walled cells are less sensitive to acute hypotonic shocks than wild-type protoplasts and *pil1Δ* walled cells, and they experience a temporary decrease in their endocytic density for about 2 min after the shock. In addition, our data allow us to estimate the relative contribution of each mechanism in the regulation of membrane tension.

The cell wall provides the largest protection during chronic and acute hypotonic shocks. Wild-type walled cells are virtually insensitive to osmotic downshifts, and *pil1Δ* walled cells are much less sensitive than *pil1Δ* protoplasts. Removal of the cell wall dramatically affects actin dynamics at endocytic sites and eisosome assembly at the plasma membrane (*Figures 1B, C* and *7* – Supplement 1B and 7 – Supplement 1C), and greatly increased the effect of hypotonic shocks on exocytosis (*Figure 4*). It is surprising that endocytosis in protoplasts still proceeds in media with osmolarity as low as 0.25 M, where a large fraction of eisosomes is disassembled. In fact, the actin endocytic machinery can overcome membrane tensions high enough to rupture the plasma membrane since we did not see stalled actin patches, or actin comet tails, in any of our experiments. Our results contrast with recent data in *S. cerevisiae* (*Riggi et al., 2019*) where endocytosis is blocked and actin comet tails are formed within 2 min of a hypotonic shock. These differences may highlight species specificities.

Our results add to a growing body of evidence that eisosomes play a critical role in the regulation of membrane tension and membrane integrity through dynamic remodeling and scaffolding of the plasma membrane (*Kabeche et al., 2015*; *Moseley, 2018*). Endocytosis in wild-type walled cells is not sensitive to chronic or acute hypotonic changes, whereas *pil1Δ* walled cells' endocytosis is (*Figure 6*). Conversely, exocytosis seems to respond more strongly to acute hypotonic shocks in wild-type walled cells than in *pil1Δ* walled cells (*Figure 4*). The protective role of eisosomes is even more striking in protoplasts under acute hypotonic shocks. Wild-type protoplasts whose plasma membrane is covered with eisosomes are largely insensitive to increases in membrane tension, whereas protoplasts with little to no eisosomes are extremely sensitive to increases in membrane tension and their plasma membrane is easily damaged (*Figure 2A–C*). Eisosomes retain this protective function even in walled cells, which becomes evident when cells are put under repeated osmolarity shocks (*Figure 2E–G*). Our micropipette aspiration experiments also demonstrate that eisosomes are critical to keep membrane tension low during an acute hypotonic shock. Therefore, our data indicate that membrane tension is decreased via the disassembly of eisosomes, through release of excess membrane surface area. Assuming eisosomes are hemi-cylinders with a diameter of ~50 nm and cells contain 1.6 μm of eisosomes per μm$^2$ of plasma membrane on average, total eisosome disassembly could release about 5% of the total surface area of the plasma membrane over ~3 min after a hypotonic shock (*Kabeche et al., 2015*), although a mild shock of $\Delta C = -0.2$ M disassembled close to ~50% eisosomes over 5 min, or about 2.5% of the surface area of the plasma membrane (*Figure 1H and I*). Single-molecule imaging in our lab demonstrated that at steady state, Pil1p undergoes rapid exchange at the eisosome ends (*Lacy et al., 2017*), potentially providing a convenient route for rapid eisosome disassembly, analogous to filament depolymerization, in combination with eisosome breaking. Disassembled eisosome components have altered phosphorylation level or

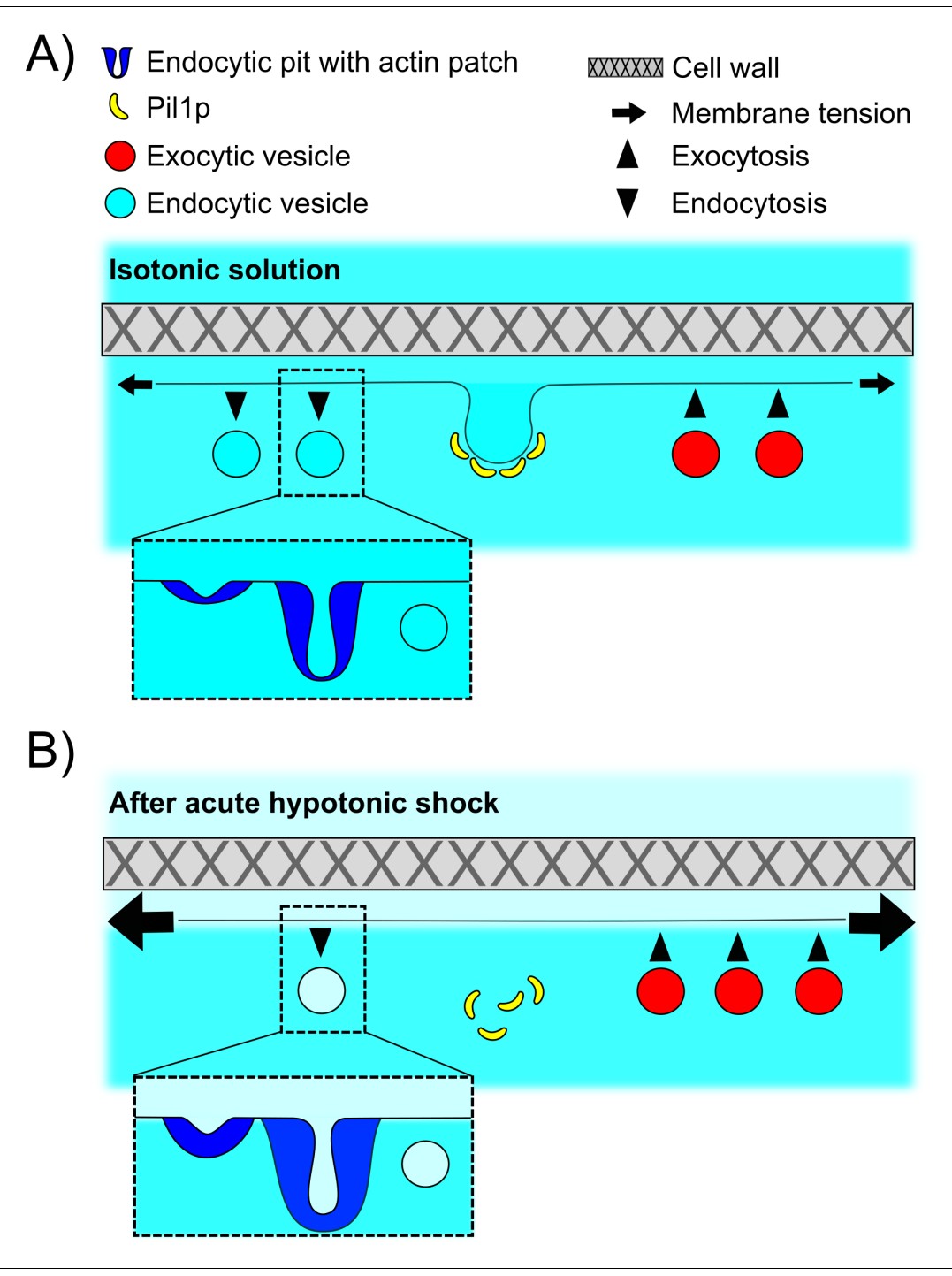

**Figure 8.** Schematic of the adaptation of fission yeast endocytosis, exocytosis, and eisosome after acute hypotonic shock-induced increase in membrane tension. (**A**) In an isotonic solution, endocytosis and exocytosis rates are largely balanced, and proteins including Pil1p are assembled at the plasma membrane to form eisosomes. Actin is recruited to endocytic sites to provide the forces needed to reshape the membrane under normal membrane tension. When present, cell wall makes fission yeast cell resistant to significant changes in the osmolarity of extracellular solution. (**B**) Acute hypotonic shock results in an increase in membrane tension, which leads to a decrease in endocytosis rate, an increase in exocytosis rate, and a rapid disassembly of eisosomes, within ~2 min. The proteins of the actin machinery are recruited in larger amounts to endocytic sites to provide larger forces for successful endocytosis under increased membrane tension. Failure of adaptation to the increase in membrane tension leads to membrane rupture and cell death in both protoplasts and walled cells.

sub-cellular localization, which potentially relays the signaling from eisosome integrity to endocytosis and/or exocytosis (*Riggi et al., 2018*; *Walther et al., 2007*), possibly via TORC2 (*Riggi et al., 2019*).

Our study highlights a third mechanism to reduce membrane tension by increasing the surface area of the plasma membrane via a temporary reduction in the endocytosis rate and an increase in the exocytosis rate. Using our data, we estimate that cells endocytose about 2% of their surface area per minute through CME, confirming our previous measurements (*Berro and Pollard, 2014a*; *Berro and Pollard, 2014b*). During acute hypotonic shocks, a reduction of the endocytosis rate plus an increase in the exocytosis rate for a few minutes would allow for a net addition of surface area to the plasma membrane. For example, in *pil1Δ* protoplasts, initially at steady state in 0.4 M sorbitol, the endocytosis rate is reduced by ~25% for ~10 min after an acute hypotonic shock of $\Delta C = -0.05$ M, while the exocytosis rate increased by ~10%. The net surface area added over that period by reduction in endocytosis and increase in exocytosis corresponds to a 5% + 6% = 11% increase in the protoplast surface area, close to the ~12% surface area increase we measured. These results confirm and quantify previous reports of control of surface tension by increasing the surface area via a modulation of endocytosis and exocytosis rates in other eukaryotes (*Apodaca, 2002*; *Homann, 1998*; *Morris and Homann, 2001*). These estimates demonstrate that modulating the endocytosis and exocytosis rates is an efficient way to increase the surface area of the plasma membrane by large amounts, but this process is relatively slow compared to eisosome disassembly. The slowness of this process might explain why *pil1Δ* and pre-stretched wild-type protoplasts that have about half the normal amount of eisosomes on their surface do not survive even relatively small hypotonic shocks, being unable to provide enough membrane in a short amount of time to reduce the tension of their plasma membrane.

For the calculations presented above, we have assumed that the size of endocytic and exocytic vesicles remain constant in all osmotic conditions. However, further experiments will be necessary to validate or invalidate this assumption. In addition, our FM4-64 data do not allow us to distinguish between lipid addition to the plasma membrane via exocytosis or via a putative transfer of lipids by non-exocytic mechanisms (*Reinisch and Prinz, 2021*), and for simplicity and by lack of further evidence, we have discussed our data as an increase in the exocytosis rate. Not much is known about lipid transfer proteins at the plasma membrane, and further experiments will be necessary to determine whether the activity of these proteins is enhanced by abrupt increases in membrane tension or hypotonic shocks.

## Molecular mechanisms driving the adaptation of the actin endocytic machinery and the rate of endocytosis under various membrane tensions

Our data demonstrate that fission yeast CME is very robust and can proceed in a wide range of osmolarities and membrane tensions. Even cells devoid of a cell wall and eisosomes were able to perform endocytosis after an acute change in membrane tension, as long as their plasma membrane was not damaged and cells remained alive. Even in the most extreme conditions tested, i.e., cells devoid of a cell wall and lacking the majority of their eisosomes, the peak number of molecules and timing of fimbrin at endocytic sites were only two times larger than those observed in wild-type walled cells.

Under conditions where membrane tension and turgor pressure were significantly increased, we observed that the endocytic actin machinery took longer and assembled a larger number of fimbrin molecules to successfully produce endocytic vesicles. This effect increased with increasing membrane tension, up to tensions high enough to rupture the cell plasma membrane. This result strongly supports the idea that the actin machinery provides the force that counteracts membrane tension and turgor pressure and deforms the plasma membrane into an endocytic pit.

The precise molecular mechanism that regulates this enhanced assembly remains to be uncovered. Our data suggest that actin dynamics is controlled via a mechanical or geometrical regulation, where actin assembles until the plasma membrane is deformed and pinched off. An alternative, and non-mutually exclusive, hypothesis is that the activity and/or recruitment of proteins upstream of the actin nucleators may be enhanced by increased membrane tension. A third hypothesis is that the decrease in the number of endocytic events after an increase in membrane tension leads to an increase in the concentration of endocytic proteins in the cytoplasm, which can then enhance the reactions performed at the endocytic sites. *Sirotkin et al., 2010* measured that 65–85% of the total

cellular content of key proteins involved in the endocytic actin machinery is localized to endocytic sites at any time. A 20% decrease in the number of endocytic sites would increase their cytoplasmic abundance by roughly 40–80%. This percentage is larger than the volume changes we measured, resulting in a net increase in the cytoplasmic concentration of these proteins, which would allow larger amounts of protein to assemble at the endocytic sites.

Conversely, the decreased endocytosis rate could be attributed to the larger number of endocytic proteins assembled at each endocytic site, which would decrease their cytoplasmic concentration. Indeed, *Burke et al., 2014* showed that modulating actin concentration modulates the number of endocytic sites in the same direction. However, it is more likely that one or several early endocytic proteins are sensitive to membrane tension, and they either fail to bind the plasma membrane or prevent the triggering of actin assembly when membrane tension is high. This idea would be consistent with results from mammalian cells demonstrating that the proportion of stalled clathrin-coated pits increases when membrane tension increases (*Ferguson et al., 2017*). In addition, several endocytic proteins that arrive before or concomitantly with the activators of the actin machinery contain BAR domains (such as Syp1p, Bzz1p, and Cdc15p), and other members of this domain family (which also includes Pil1p) have been shown to bind membranes in a tension-sensitive manner. Further quantitative studies of early endocytic proteins will help uncover the validity and relative contributions of each one of these hypotheses.

We expect our results to be relevant to the study of CME and membrane tension regulation in other eukaryotes since the molecular machineries for endocytosis, exocytosis, and osmotic response are highly conserved. In addition, regulation of membrane tension and CME are particularly critical during cell polarization (*Mostov et al., 2000*), during neuron development and shape changes (*Urbina et al., 2018*), and at synapses where large pools of membranes are added and retrieved on a very fast time scale (*Nicholson-Fish et al., 2016*; *Watanabe and Boucrot, 2017*).

# Materials and methods

**Key resources table**

| Reagent type (species) or resource | Designation | Source or reference | Identifiers | Additional information |
|---|---|---|---|---|
| Strain, strain background (*Schizosacchoromyces pombe*) | Fim1p-mEGFP | *Berro and Pollard, 2014a* | SpJB57 | fim1-mEGFP-NatMX6 ade6-M216 his3-Δ1 leu1-32 ura4-Δ18 h+ |
| Strain, strain background (*Schizosacchoromyces pombe*) | Pil1p-mEGFP | *Lacy et al., 2017* | SpJB204 | pil1-mEGFP-kanMX6 ade6-M216 his3-Δ1 leu1-32 ura4-Δ18 h- |
| Strain, strain background (*Schizosacchoromyces pombe*) | *pil1Δ* Fim1p-mEGFP | This study | SpJB234 | pil1Δ fim1-mEGFP-NatMX6 ade6-M216 his3-Δ1 leu1-32 ura4-Δ18 h- |
| Strain, strain background (*Schizosacchoromyces pombe*) | mScarlet-I-End4p mEGFP-Fim1p | This study | SpJB566 | mScarlet-I-end4 mEGFP-fim1 fex1Δ fex2Δ ade6-M216 his3-Δ1 leu1-32 ura4-Δ18 h- |
| Chemical compound, drug | FM4-64 | Biotium, Fremont, CA | | |
| Chemical compound, drug | Sulforhodamine B | MP Biomedicals LLC, Santa Ana | | |
| Chemical compound, drug | Latrunculin A | Millipore, MA | | |
| Chemical compound, drug | Brefeldin A | Santa Cruz Biotechnology Inc, TX | | |
| Software, algorithm | Matlab | Mathworks | R2016a, R2018b | |

*Continued on next page*

*Continued*

| Reagent type (species) or resource | Designation | Source or reference | Identifiers | Additional information |
|---|---|---|---|---|
| Software, algorithm | FIJI ImageJ | *Schindelin et al., 2012*; *Schneider et al., 2012* | | |
| Software, algorithm | Trackmate | *Tinevez et al., 2017* | | |
| Software, algorithm | PatchTrackingTools | This study, *Berro, 2018* | Version 2015.12.16 | https://bitbucket.org/jberro/patchtrackingtools/src/stable/, copy archived at swh:1:rev: 226505c08c21 97584c1299462b5de85433d1fcf1. |
| Software, algorithm | PatchTrackingData Postprocessing. Deterministic | This study, *Berro, 2021* | #1B100-4 | https://bitbucket.org/jberro/patchtrackingdatapostprocessing.deterministic/src/master/, copy archived at swh:1:rev:3d96cd606f47a692d d6db55ba2aab0a7f6e0c4f2. |
| Other | Glass micropipette | World Precision Instruments, Sarasota | Version 160801 | |
| Chemical compound, drug | β-casein | Millipore-Sigma | Saint-Louis | |
| Software, algorithm | GraphPad Prism | Software, La Jolla, CA | version 8.4.2 | |

## Yeast strains and media

The *S. pombe* strains used in this study are listed in the Key Resources Table. Yeast cells were grown in YE5S (yeast extract supplemented with 0.225 g/l of uracil, lysine, histidine, adenine, and leucine), which was supplemented with 0–1.2 M D-sorbitol, at 32°C in the exponential phase for about 18 hr. Cells were washed twice and resuspended in filtered EMM5S (Edinburgh Minimum Media supplemented with 0.225 g/l of uracil, lysine, histidine, adenine, and leucine), which was supplemented with the same concentration of D-sorbitol, at least 10 min before imaging so they can adapt and reach steady state.

## Protoplast preparation

*S. pombe* cells were grown in YE5S at 32°C in exponential phase for about 18 hr. 10 ml of cells were harvested and washed two times with SCS buffer (20 mM citrate buffer, 1 M D-sorbitol, pH = 5.8), and resuspended in SCS supplemented with 0.1 g/ml Lallzyme (Lallemand, Montreal, Canada) (*Flor-Parra et al., 2014*). Cells were incubated with gentle shaking for 10 min at 37°C in the dark except for experiments in *Figure 5*, where cells were digested at room temperature with gentle shaking for 30 min in the presence of inhibitors. The resulting protoplasts were gently washed twice in EMM5S with 0.25–1.2 M D-sorbitol, spun down for 3 min at 960 rcf between washes, and resuspended in EMM5S buffer supplemented with 0.25–1.2 M D-sorbitol at least 10 min before imaging so that they can adapt and reach steady state.

## Microscopy

Microscopy was performed using a spinning disk confocal microscope, built on a TiE inverted microscope (Nikon, Tokyo, Japan), equipped with a CSU-W1 spinning head (Yokogawa Electric Corporation, Tokyo, Japan), a x100/1.45NA phase objective, an iXon Ultra888 EMCCD camera (Andor, Belfast, UK), and the NIS-Elements software v. 4.30.02 (Nikon, Tokyo, Japan). The full system was switched on at least 45 min prior to any experiment to stabilize the laser power and the room temperature. Cells were loaded into commercially available microfluidic chambers for haploid yeast cells (Y04C-02-5PK; Millipore-Sigma, Saint-Louis) for the CellASIC ONIX2 microfluidics system (Millipore-Sigma). Each field of view was imaged for 60 s, and each second, a stack of six z-slices separated by

0.5 µm was imaged. The microscope was focused such that the part of the cell closest to the coverslip was captured.

## Acute hypotonic shocks

Walled cells or protoplasts were first imaged in their steady-state media (EMM5S supplemented with 0–1.2 M D-sorbitol). The steady-state media were exchanged with media supplemented with a lower D-sorbitol concentration (the concentration difference is noted, ΔC), with an inlet pressure of 5 psi. These hypotonic shock media were labeled with 6.7 µg/ml of sulforhodamine B (MP Biomedicals LLC, Santa Ana), a red cell-impermeable dye that allowed us to (a) monitor the full exchange of the solution in the microfluidic chamber prior to image acquisition and (b) monitor the plasma membrane integrity of the cells after the shock. In each condition, the first movie was started when the sulforhodamine B dye was visible in the field of view. For clarity, this time point is labeled t = 0 min in all our figures, but note that we estimate it may vary by up to ~30 s between movies and conditions. We imaged cells by taking one stack of six z-slices per second for 60 s. After the end of each movie, we rapidly changed field of view and restarted acquisition 1 min after the end of the previous movie, so that movies started every 2 min after the acute hypotonic shock. Tracks from cells that contained red fluorescence from the sulforhodamine B dye were excluded from the analysis, because this indicated that cell membrane had been damaged.

## Inhibition of endocytosis and exocytosis during acute hypotonic shocks

Endocytosis or exocytosis was inhibited by including, respectively, 25 µM Latrunculin A (Millipore, MA) or 2 mM Brefeldin A (Santa Cruz Biotechnology Inc, TX) in the solution used to prepare the protoplasts and to perform the hypotonic shocks. Hypotonic shock solution also included 20 µM FM4-64 (Biotium, Fremont, CA) to stain dead protoplasts (*Vida and Gerhardt, 1999*; *Figure 5A*), and inlet pressure was set at 4 psi.

## Measurement of the temporal evolution of the number of proteins and speed

Movies were processed and analyzed using an updated version of the PatchTrackingTools toolset for the Fiji (*Schindelin et al., 2012*) distribution of ImageJ (*Berro and Pollard, 2014a*; *Schneider et al., 2012*). This new version includes automatic patch-tracking capabilities based on the Trackmate library (*Tinevez et al., 2017*) and is available on the Berro lab website http://campuspress.yale.edu/berrolab/publications/software/ and the lab bitbucket https://bitbucket.org/jberro/. Prior to any quantitative measurement, we corrected our movies for uneven illumination and camera noise. The uneven illumination was measured by imaging a solution of Alexa 488 dye, and the camera noise was measured by imaging a field of view with 0% laser power. We tracked Fim1-mEGFP spots with a circular 7-pixel diameter region of interest (ROI), and measured the temporal evolution of the fluorescence intensities and the position of the centers of mass. The spot intensity was corrected for cytoplasmic background using a 9-pixel median filter and was then corrected for photobleaching. The photobleaching rate was estimated by fitting a single exponential to the temporal evolution of the intensity of cytoplasmic ROIs void of any identifiable spots of fluorescence (*Berro and Pollard, 2014a*). Only tracks longer than 5 s and displaying an increase followed by a decrease in intensity were kept for the analysis. Individual tracks were aligned and averaged with the temporal super-resolution algorithm from *Berro and Pollard, 2014a* and post-processed to generate figures using the Matlab R2016a (Mathworks) scripts PatchTrackingDataPostprocessing.Deterministic, available on the Berro lab website http://campuspress.yale.edu/berrolab/publications/software/ and the lab bitbucket https://bitbucket.org/jberro/. In brief, this method realigns temporal signals that have low temporal resolution and where no absolute time reference is available to align them relatively to each other. It iteratively finds the temporal offset, which has a higher precision than the measured signal and minimizes the mean square difference between each measured signal and a reference signal. For the first round of alignments, the reference signal was one of the measurements. After each realignment round, a new reference was calculated as the mean of all the realigned signals, which is an estimator of the true underlying signal.

To control and calibrate the intensity of our measurements, we imaged wild-type walled cells expressing Fim1p-mEGP each imaging day. Intensities were converted into number of molecules

with a calibration factor such that the peak intensity of our control strain corresponded to 830 molecules (*Berro and Pollard, 2014a*).

In all figures presenting the temporal evolution of the number of molecules or the speed, time 0 s corresponds to the time point when the number of molecules is maximum (also called the peak number). The 'speed vs time' plots help determine whether endocytosis completes normally in different conditions. At a given time point, the speed corresponds to the average movement of the endocytic structure in the plane of the membrane between two consecutive images. For an endocytic event that completes normally, the speed is close to 0 while the endocytic pit elongates before the vesicle is pinched off. Note that the speed is not exactly 0 because of localization errors and putative small movements of the endocytic structure in the plane of the membrane. The speed increases after the vesicle is pinched off and it diffuses freely in the cytoplasm. Since this movement is mostly diffusive, the standard deviations of the speeds are large.

Statistical tests between conditions were performed at time 0 s with a one-way ANOVA test using the number of tracks collected to build the figure. To avoid extrapolating the data, we compared the relative duration of assembly and disassembly between conditions using the time at which the average number of molecules reaches half the peak number.

## Measurement of the density of CME events

We used the *S. pombe* profiling tools for ImageJ (*Berro and Pollard, 2014a*) to measure the number of endocytic events at a given time in each cell. In brief, on a sum-projected z-stack, we manually outlined individual cells, and, for each position along the long axis of a cell, we measured the sum of fluorescence orthogonal to the long axis. We corrected the intensity profile in each cell for its cytoplasmic intensity and media fluorescence outside the cell. We estimated the number of patches in each cell by dividing the corrected fluorescence signal with the temporal average of the fluorescence intensity of one endocytic event. We calculated the linear density of endocytic events as the ratio between the number of endocytic events in a cell and its length.

We estimated the percentage of plasma membrane internalized by endocytosis per minute as follows. We measured ~120 endocytic sites per cell at a given time on average. Since the actin meshwork assembles and disassembles in about 20 s, we estimated that 360 endocytic vesicles are formed per minute. Assuming endocytic vesicles have a diameter of 50 nm, this corresponds to an ~2.8 $\mu m^2$ area endocytosed per minute or ~2% of the total surface area, considering the average cell is around 12 $\mu m$ long and has a 132 $\mu m^2$ area of plasma membrane (assuming the cell shape is a cylinder capped by two hemispheres).

## Measurement of the exocytosis rate with FM4-64 staining

The exocytosis rate was measured by combining the acute hypotonic shock with FM4-64 staining, in a similar approach as has been reported (*Gauthier et al., 2009*; *Smith and Betz, 1996*; *Vida and Emr, 1995*). The cell impermeable dye FM4-64 (Biotium, Fremont, CA) was diluted to a final concentration of 20 $\mu M$ in any of the media used. When cells are exposed to FM4-64, the dye rapidly stains the outer leaflet of the plasma membrane. Upon endocytosis, the dye is trafficked inside the cell without change in fluorescence. The total cell fluorescence intensity was measured after segmenting the cells by thresholding the fluorescence signal above background levels. The fluorescence intensity was normalized to the intensity reached at the end of the fast increase ~1 min after the dye was flowed in, which corresponds to the intensity of total surface area of the plasma membrane (*Figure 4B*). After this fast phase (<20 s), the fluorescence signal increased more slowly every time the unstained membrane was exposed to the cell surface by exocytosis. At a short time scale (~5–20 min depending on the exocytosis rate), recycling of stained membrane is negligible and one can assume that all exocytosed membrane is virtually unstained. Since the intensity at the beginning of the slow phase was normalized to 1, the slope of the linear increase of fluorescence is equal to the amount of membrane exocytosed per minute, expressed as a fraction of the surface area of the plasma membrane. For all measurements, images were taken at 5 s intervals at the middle plane of cells with the help of Perfect Focusing System (Nikon, Tokyo, Japan), with minimal laser excitation in order to reduce toxicity and photobleaching to negligible values. Curve fitting and slope calculation were performed in GraphPad Prism (GraphPad Software, La Jolla, CA).

## Measurement of eisosomes' density on the plasma membrane

We imaged full cells expressing Pil1p-mEGFP by taking stacks of 0.5-µm-spaced z-slices. We corrected these z-stacks for uneven illumination and manually outlined individual cells to determine the surface area of each cell. To determine the total amount of eisosome-bound Pil1p-mEGFP, we subtracted the cytosolic intensity of Pil1-mEGFP using a pre-determined threshold and summed all the z-slices. We measured the mean membrane intensity of each cell on the thresholded sum-projection image. The eisosome density was determined by dividing this mean intensity by the surface area of each protoplast.

To quantify the relative changes in area fraction of eisosomes after acute hypotonic shocks, wild-type protoplasts expressing Pil1p-mEGFP were loaded into ONIX2 microfluidics system (Millipore-Sigma, Saint-Louis), and time-lapse fluorescent images were taken at a single z-slice at the top of protoplasts during media change. After background correction, the total area fraction of eisosomes at the beginning of the hypotonic shock was set to 1.0 for normalization, and the normalized values of area fraction were fit to a single exponential decay curve in GraphPad Prism (GraphPad Software; La Jolla, CA).

## Measurement of membrane tension

Protoplasts were loaded in a custom-built chamber, which was passivated with 0.2 mg/ml β-casein (Millipore-Sigma, Saint-Louis) for 30 min and pre-equilibrated with EMM5S supplemented with 0.8 M D-sorbitol. A glass micropipette (#1B100-4; World Precision Instruments, Sarasota) was forged to a diameter smaller than the average protoplast radius (~2.5 µm) and was connected to a water reservoir of adjustable height to apply a defined aspiration pressure. Before and after each experiment, the height of the water reservoir was adjusted to set the aspiration pressure to 0. Cells were imaged with a bright-field IX-71 inverted microscope (Olympus, Tokyo, Japan) equipped with a 60x/1.4NA objective, and images were recorded every second. Aspiration pressure was gradually increased every 30 s and the membrane tension $\sigma$ was calculated as $\sigma = P.R_p / \left[2\left(1 - R_p/R_c\right)\right]$, where $R_p$ and $R_c$ are, respectively, the micropipette and the cell radius, $P$ is the aspiration pressure for which the length of the tongue of the protoplast in the micropipette is equal to $R_p$ (*Evans and Yeung, 1989*). To limit the effects of the adaptation of cells' membrane tension, all measurements were performed within the first 5 min after the hypotonic shock, which greatly limited the throughput of our assay (one measurement per sample), compared to the measurements at steady state (around six measurements per sample).

## Acknowledgements

We thank Yale West Campus Imaging core for providing access to the spinning disc confocal microscope, members of the Berro lab for insightful discussions, and R Fernandez for providing mutant yeast strains. We thank Ramesh Ramji and Kathryn Miller-Jensen for initial help with the microfluidics part of this project. We gratefully thank Millipore Sigma for lending us a CellASIC unit and Erdem Karatekin for allowing access to his micropipette aspiration setup. We also thank Samantha Dundon, Mike Lacy, and Matt Akamatsu for their comments on our manuscript. This research was supported in part by National Institutes of Health/National Institute of General Medical Sciences Grant R01GM115636 and by seed funding from the American Cancer Society Institutional Research Grant #IRG 58-012-58.

## Additional information

### Funding

| Funder | Grant reference number | Author |
| --- | --- | --- |
| National Institute of General Medical Sciences | R01GM115636 | Julien Berro |
| American Cancer Society | IRG 58-012-58 | Julien Berro |

The funders had no role in study design, data collection and interpretation, or the decision to submit the work for publication.

### Author contributions
Joël Lemière, Yuan Ren, Conceptualization, Data curation, Formal analysis, Investigation, Methodology, Writing - original draft, Writing - review and editing; Julien Berro, Conceptualization, Data curation, Software, Formal analysis, Supervision, Funding acquisition, Investigation, Methodology, Writing - original draft, Writing - review and editing

### Author ORCIDs
Joël Lemière  https://orcid.org/0000-0002-9017-1959
Yuan Ren  https://orcid.org/0000-0001-7155-7664
Julien Berro  https://orcid.org/0000-0002-9560-8646

### Decision letter and Author response
Decision letter https://doi.org/10.7554/eLife.62084.sa1
Author response https://doi.org/10.7554/eLife.62084.sa2

## Additional files

### Supplementary files
• Supplementary file 1. Number of cells used to generate the figures of this paper. (**a**) *Figure 3A* and *Figure 3B*. (**b**) *Figure 3C*. (**c**) *Figure 3D*. (**d**) *Figure 6C* and *Figure 6—figure supplement 2A*. (**e**) *Figure 6D*. (**f**) *Figure 6E*. (**g**) *Figure 6G*. (**h**) *Figure 6H*. (**i**) *Figure 7C* and *Figure 7F*. (**j**) *Figure 7D* and *Figure 7G*. (**k**) *Figure 7H* and *Figure 7I*. (**l**) *Figure 7—figure supplement 1B* and *Figure 7—figure supplement 1C*.

• Transparent reporting form

### Data availability
All data generated or analysed during this study are included in the manuscript and supporting files.

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
