## [Decision Letter]

[Editors’ note: the authors submitted for reconsideration following the decision after peer review. What follows is the decision letter after the first round of review.]

Thank you for submitting your work entitled "Adaptation of actin dynamics and membrane tension control for yeast endocytosis" for consideration by *eLife*. Your article has been reviewed by 3 peer reviewers, one of whom is a member of our Board of Reviewing Editors, and the evaluation has been overseen by a Senior Editor. The following individual involved in review of your submission has agreed to reveal their identity: Aurélien Roux (Reviewer #3).

Our decision has been reached after consultation between the reviewers. Based on these discussions and the individual reviews below, we regret to inform you that your work will not be considered further for publication in *eLife*.

The reviewers concurred on the quality and the overall significance of your data. However, the paper appears to be composed of two disjointed stories that are not mature yet. Considering the effort in terms of additional experiments required to present a conceptual advance, we regrettably cannot move forward with the paper at *eLife*. We encourage you to resubmit a new version that contains the data requested or you may choose to submit this work elsewhere for its timely publication.

*Reviewer #1:*

This study is of interest as the role of eisosomes in fission yeast remains poorly explored and the findings presented by Lemière and Berro are likely to be relevant for the study of CME and its relation with membrane tension in mammalian cells. Several controls need however to be performed to better support the conclusions of this otherwise timely contribution.

1. My main concern has to do with the readout used in this study. The authors assume that following the dynamics of fimbrin allow to monitor endocytosis and actin dynamics. What is the evidence that fimbrin molecules are recruited to endocytic patches? Several control experiments should be performed to validate this claim including clathrin colocalization to fimbrin patches. In addition, inhibiting CME (dyn mutant or else) should then result in decrease of fimbrin recruitment. Several of the effects of osmotic shocks described on fimbrin dynamics could occur independently from CME and the authors ought to correlate these effect with cargo uptake. What is the effect of fimbrin deletion on endocytosis? As for actin dynamics, why did not the authors follow actin directly? Following the dynamics of actin or another actin binding protein would reinforce this aspect of the work.

2. I am somehow concerned by the degree of novelty of this study as it has long been known in mammalian cells that membrane tension controls the rate of endocytosis and exocytosis (see Sheetz seminal papers and more recently papers such as Boulant et al.,). Moreover, eisosomes have been recently established as membrane reservoirs in a study with similar data in yeast (protective role of the wall and disappearance of eisosomes through flattening under increased osmolarity in protoplasts as decribed in Kabeche, JCS 2015). As a matter of facts, these published data should be better acknowledged in the results and discussion. For instance, on line 178, the authors write that their data "suggest that eisosomes might be disassembled to reduce membrane tension in media with low osmolarity…" this has been already demonstrated in Kabeche, JCS 2015. In this context, the new finding reported in the current study is mostly about the role of endocytosis on membrane tension, that is, that endocytosis is transiently reduced to decrease membrane tension. However, in none of the presented experiments have the authors addressed the selective contribution of endocytosis. The authors should therefore re-examine their findings when endocytosis is blocked by drugs or mutants. Thus, measuring membrane tension variations under osmotic shock when endocytosis is blocked would allow to estimate the relative contribution of endocytosis in membrane tension buffering.

3. The absence of eisosomes should be documented in Pil1∆ mutants. Line 244: what is the evidence that WT protoplasts express normal amounts of Pil1p when they assemble fewer eisosomes? Data presented in Figure 6 should be correlated with the visualization of eisosomes as in Figure 4B.

*Reviewer #2:*

Lemiere et al., use quantitative fluorescence microscopy in the model organism fission yeast to address the complex relationship between cellular processes such as endocytosis and cellular structures such as eisosomes to regulate and equilibrate plasma membrane tension. Dissecting individual processes that mechano and/or biochemically regulate plasma membrane tension remains to be challenging. Single cellular organisms such as fission yeast therefore provide a versatile and easy to manipulate model system to tackle these particular very important regulatory processes and can translate crucial findings to higher eukaryotic model systems.

The authors use their previously established quantitative fluorescence microscopy and analytical approach to measure the response of the actin crosslinker fimbrin to changes in the number of molecules, lifetimes and speed to osmolarity media changes that are meant to change plasma membrane tension. The authors describe in a combination of experiments conducted in walled cells and protoblasts, both also mutated for the Pil1 eisosome component, and changed dynamics of fimbrin. These findings provide very exciting new results about how the actin machinery adapts to osmolarity changes. These are even combined with membrane tension measurements. Together with determining the density of endocytic events after hypotonic shock treatment, the authors conclude that the cooperation of eisosome disassembly, decrease of endocytic events and ongoing exocytosis are responsible for equilibrating plasma membrane tension changes.

The findings in the manuscript are of descriptive nature and do not identify mechno-regulatory or biochemical mechanisms that control the membrane reservoir or the adaption of actin dynamics in response to changing media osmolarities. Our major concern is that the overall main point of the paper is not clear. It seems that the authors present two very interesting half stories that have not yet have been completed and that do not stand on their own in a complete story. Given the allowed time frame of revision, it is not clear that the authors could make adequate revisions. If they can, we would suggest that they concentrate on one of their respective stories and endeavor to provide more mechanistic insights into either "the adaption of actin dynamics in yeast endocytosis in response to membrane tension changes" or "cellular mechanisms that control the plasma membrane reservoir in response to osmolarity changes". Detailed comments can be found below.

1. As mentioned above, it seems that the authors pursued two different stories in this manuscript and tried to cover them under one main point. This creates confusion for the reader throughout the manuscript especially in Figure 4 and Figure 5.

Therefore, we suggest that the authors concentrate on one of the two take subjects that they address. The two different stories that could be expanded for a manuscript are:

a. One on the adaptive dynamics of the actin crosslinker fimbrin. The authors precisely report on the number of molecules, lifetimes and speed changes that happen under hypotonic shock treatments.

However, the authors do not pursue how actin dynamics mechanistically adapt to hypotonic shock treatments. Fimbrin is a crosslinker and there are many other interesting candidate proteins whose dynamic behavior under hypotonic shock treatments the authors could explore. Thus, it would be possible for the authors to obtain precise mechanistic details about how "actin dynamics adapt to membrane tension increase to control yeast endocytosis". Previously published work of the last author shows that these strains are available. We would like to see experiments equivalent to the one conducted with the fimbrin-FP expressing yeast strain with Sla1, Arp2/3 and an early endocytic marker. If the authors are willing to add these experiments, we would also like to see the measurements of membrane tension of the protoblasts treated in the conditions used in Figure 5. A direct relationship between actin cytoskeleton protein binding dynamics and quantified membrane tension is of high interest for the endocytic community and would make this manuscript very valuable and significantly new. Then the authors could explain and discuss the differences they see in fimbrin dynamics, which they do not include on in the main text. In Figure 2E and Figure3D, fimbrin lifetimes are faster under hypotonic treatment. However, in Figure 5, lifetimes decrease under hypotonic treatment. We do not understand what the authors are looking at in these analyses. Determining membrane tension for all experimental conditions is essential for the correct interpretation of these results (of course, this is only possible for protoplasts). In addition, the authors should think about showing experimentally how faster or slower fimbroin dynamics can be related to productive endocytic events.

b. The other main point the authors pursue is the regulatory mechanism for membrane tension equilibration in response to osmolarity changes. Here the authors suggest focus on these regulatory components: the endocytic machinery, eisosomes, the cell wall and the exocytic machinery. With their experiments the authors only cover the dynamic behavior of the actin crosslinker fimbrin as the representative marker for the dynamics of endocytic events in response to hypotonic treatment and the observation that PIL1 deleted cells are more prone to hypotonic shock.

What follows are conclusions made by the authors despite a lack of experimental evidence:

- 'eisosomes play a critical role in the regulation of membrane tension and membrane integrity through dynamic remodeling and scaffolding of the plasma membrane' p9, line 342.

- the 'protective role of eisosomes is even more striking in protoplasts under acute hypotonic shocks'.

- 'our data indicate that membrane tension is decreased via the disassembly of eisosomes, through release of excess membrane surface area.' p9, line 352.

- Lines 361 – 367: 'Our study identified a third, new mechanism to reduce membrane tension by temporarily reducing the endocytic density. Using our data and previous measurements (18,19), we estimate that cells endocytose about 2% of their surface area per minute. Assuming that at steady state the surface area added to the plasma membrane via exocytosis balances the surface area that is endocytosed, a reduction of the endocytic rate while keeping the exocytic rate constant for a few minutes would allow for a net addition of surface area to the plasma membrane'.

The authors only show images of eisosomes at steady state e.g. Figure 4a,b,c. To investigate the regulatory mechanism and provide convincing evidence for their claims, we suggest that the authors also track Pil1 dynamics under different hypotonic treatments and the dynamics of an exocytic marker. Here again, a direct relation to membrane tension values measured under the experimental conditions used would be crucial.

2. The title of the manuscript, "Adaption of actin dynamics and membrane tension control for yeast endocytosis," does not describe the work well since the authors only perform experiments on the actin crosslinker fimbrin and quantification of membrane tension was only performed for one experimental condition. We suggest that the authors choose a more specific title that states the main point of the revised manuscript.

3. "Temporal super-resolution" is not yet a common method that the reader should be expected to know about. Moreover, the reader might not be familiar with the improvements of this method performed by the authors (Line 98). We suggest that they schematically describe the method of temporal super-resolution in a supplementary Figure to Figure 1 and also explain the improvements. It is also not clear in which plane the cells are imaged and why only a subset of fimbrin tracks where kept for the further analysis (see comment 4.)

4. It would be helpful if the authors would provide a thorough analysis of how many fimbrin tracks there are in total, what classes of behavior can be seen and on which subset the authors are focusing and why.

The authors state in their methods, that all tracks were discarded that did not match the following criterion: 'longer than 5s and displaying an increase followed by a decrease in intensity' p12 line 497. This introduces a significant bias towards productive tracks. Tracks that are stalled or last longer than the observation window would not enter the analysis.

Why did the authors did not discard tracks that 'failed to proceed normally, as reported by the virtually null speed' p6 line 196? Or did they use a different criterion for these tracks?

5. Quality of Figures: Overall the plot quality could be improved by reporting the confidence interval differently. The plots become very busy and the lines overlap strongly, so that the reader has to look at the plots with a higher zoom to differentiate between the actual data lines and the lines of the confidence interval. We suggest that they plot the lifetimes of endocytic events separately and report statistical tests on the lifetime distributions. The way the Figure legends are placed in Figure 4 is very confusing and can be improved in the other figures for clarity, as well.

6. A model Figure summarizing their main conclusions at the end of the manuscript would clarify the take home message for the reader. The summary could make points about: What factors influence membrane tension control? Or, they could provide a schematic representation for how the actin machinery adapts to increases in membrane tension.

7. The plots in Figure 5H and I and the conclusion in the text about the Pil1 cytoplasmatic pool are very confusing and would be better placed in a supplementary Figure.

*Reviewer #3:*

The paper by Lemiere and Berro focuses on the dynamics of endocytic events and eisosomes in walled cells and protoplasts, to investigate the role of membrane tension in both endocytic bud formation and eisosomes. It builds on previous work that has shown that clathrin endocytosis dnamics is affected by changes in membrane tension, and that eisosomes participate into the regulation of plasma membrane tension. The paper is clearly written, and has interesting findings, among which are:

1. That membrane tension changes do not strongly affect CME in walled cells

2. Embrane tension strongly affect CME in protoplasts

3. Eisosomes, while having a limited role in walled cells, have a strong role in regulating membrane tension in protoplasts.

I have a few concerns about the paper in the current state. The authors want to compare WT (walled) cells and protoplasts, but the comparison is not done thoroughly, which makes the paper a bit unbalanced. Also, the fact that the paper is written in two parts, first the walled cell results, and then the protoplasts results, the comparison is not very easy to follow. The authors may reconsider the text organization by clustering similar experiments done in wall cells and protoplasts. I understand that some experiments cannot be performed in protoplasts or in walled cells, but the following comments are here to extend as much as possible the comparison.

– The density of eisosomes should be better characterized in walled cells and in protoplasts, before and after the shock (see Figure 4B and 4C, if I understood well, this is only for cells at steady state, and different osmotic pressure). The authors model proposes that eisosomes regulate tension through disassembly. The authors have to show this. I would recommend to also perform hypertonic shocks that should increase the number of eisosomes.

– Generally speaking, it sounds strange that eisosomes would have an impact only on protoplast. The observation that when grown at higher osmotic pressure, cells have more eisosomes suggest that cells prepare to stringer hypo-osmotic shocks in this condition. How can the author rationalize these results?

– I was wondering how sure the Fim1p reporter reports actin density? I guess this has been published elsewhere as the authors rely on published work on this point, but it would still be nice to have a figure panel that shows a co-labelling with actin.

– I find surprising that the dynamics of actin is not different upon osmotic shocks in WT walled cells. I was wondering if the actin treadmilling was modified upon osmotic shocks. This should be easy to test through FRAP experiment of Fim1p at the endocytic sites.

– Reported tension values are extremely small (10-4 nM/um, which is 10-7 N/m). Fibroblastic cells have tensions of typically 10-4 – 10-5 N/m. I expect yeast cells to have even higher tension become they live in higher turgor pressure. Also at 10-7 N/m, membrane usually have large fluctuations which are visible by any type of microscopy, which are not seen in the authors’ experiments. Either the calculation has a problem, or the unit is wrong. Please use standard units for membrane tension in N/m or in mN/m (or m-1).

[Editors’ note: further revisions were suggested prior to acceptance, as described below.]

Thank you for resubmitting your work entitled "Rapid adaptation of endocytosis, exocytosis and eisosomes after an acute increase in membrane tension in yeast cells" for further consideration by *eLife*. Your revised article has been evaluated by Vivek Malhotra (Senior Editor) and a Reviewing Editor.

The manuscript has been improved but there are some remaining issues that need to be addressed before acceptance, as outlined below:

As you will see in the verbatim comments attached, the first reviewer agrees that the revised manuscript contains a substantial amount of work and that it has increased in length. While Reviewer 1 acknowledges that some aspects of the manuscript have greatly improved, the reviewer is still concerned by the readability of the manuscript, a concern raised already for the initial submission. This is exacerbated by the addition of a third part on exocytosis. In particular, the reviewer finds it difficult to follow the comparison between walled cells and protoplasts. The reviewer expresses some concerns about the readability of the plots expressed as "speed vs time". The reviewer also disagrees with the use of deltaP for accounting for osmolarity changes. Finally, related to the new part on exocytosis, the reviewer is asking for a better characterization of its regulation by osmotic shock.

Reviewer 2 is somehow less critical and agrees that the manuscript strength resides in the highly quantitative characterization of endocytosis dynamics and in the study of the 3 co-operating mechanisms at play during the response to hypo-osmotic shock.

Considering the positive appreciation of your work by the two reviewers, I am willing to leave the door open for a revised version. Since this work is a resubmission of a manuscript already reviewed in 2018, we will be very strict on the revised manuscript. It should comply with all of the 2 reviewers concerns with a special attention to the readability of the manuscript. I would like to insist that another round of reviewing will not be granted at this stage.

*Reviewer #1:*

This submission is a revised version of a previous submission. The authors have done a substantial amount of work, and the paper has increased in length. Some aspects have greatly improved, such as the characterization of the roles of the cell wall, tension and actin (using drugs and mutants to change actin organization) in controlling endocytosis. The role of eisosomes in buffering membrane tension has been characterized in further details, providing the first evidence that eisosomes could play a similar role in yeast than caveolae in cells in buffering membrane tension. Finally, the authors added a study on the evolution of exocytosis during hypoosmotic shocks, which reacts in opposite direction than endocytosis.

While I value the efforts and work provided by the authors in their revised version, they in fact took an opposite direction in their revised manuscript than what was initially suggested by the reviewers: – split the manuscript in two, and strengthen the characterization of tension/endocytosis on the one hand, and eisosomes/tension on the other. Instead, the authors kept the two parts in the same manuscript, and added a third part on exocytosis. They also extended their comparison of walled cells and protoplasts, which is indeed interesting, but makes the flow hard to follow because the reader needs to go back-and-forth between different figures. This choice is at the expense of the manuscript's readability, which became lengthy and difficult to follow.

While I value the efforts made by the authors, and the improvement of the characterization of endocytosis, tension and eisosomes, I am embarrassed to say that the main criticism remains. I understand that the authors wanted to broaden their study, and get more general conclusions, but it does not go through well. I unfortunately cannot recommend publication in the present form, and would certainly suggest again to split the manuscript in two (which could be back-to-back in *ELife* for example), one on endocytosis/exocytosis and tension (where pil1delta and protoplasts would be used as a tool to dysregulate tension) and one on eisosomes/tension buffering.

– While the mechanism at play to regulate dynamics of endocytosis and eisosomes during osmotic shock became well characterized, the new part on exocytosis is very short and does not provide any further understanding of the regulation at play here. Is it tethering, or SNARE function that is regulated by tension? Is that higher tension strengthens the actin dome around exocytic vesicles to trigger fusion? A better characterization, at least at the same level of other parts would be needed here.

– Some plots are difficult to read. The plot "Speed" vs "time", which zero the invagination time, is not easy to read. The total duration of endocytic events is changing with conditions, and if I understood well, that is the main effect that the author look at, but it took me quite some time to figure out how to read these. I think a classic plot with average duration would be easier to follow for most readers. I understand that in some cases, the authors see no invagination, which is also reported in this graph, but I think 2 plots would do a better job than a single one in this case. Or perhaps, explain better how to read those plots, as they are uncommon, and at the basis of the authors' argumentation.

– I disagree with the use of deltaP to account for a change of osmolarity. Pressure should be in Pa (or any corresponding unit), not in Moles per Liter. Either the authors use deltaC, or they measure the osmotic pressure of their solution and can use Osm in this case for pressure. Also, at concentrations about 0.5M, there is usually a substantial discrepancy between Osmolarity and Molarity, and as the authors are using Molarities above 1M, I think it would be safer to estimate the osmotic pressure of those solutions.

*Reviewer #2:*

Lemiere and colleagues show in this manuscript that endocytosis is remarkably robust against osmotic changes in normal walled yeast cells. However, in cells without cell walls endocytosis is much more sensitive to osmotic changes, presumably due to changing membrane tension. Eisosomes are shown to be important for regulating membrane tension after osmotic shocks.

Changes in turgor pressure are also shown to transiently affect the frequency of endocytic events. Eisosomes are shown to be important for restoring the normal endocytic frequency after shocks. Finally, exocytosis is suggested to be a mechanism that restores normal membrane tension after hypotonic shocks.

The strength of the manuscript is the highly quantitative and careful experimentation and the fact that three different plasma membrane processes are considered together (endocytosis, exocytosis, eisosomes), which is surprisingly rarely done, even though they must be functionally connected.

I have some suggestions for the authors to improve their manuscript:

Lines 217-223: Does the plasma membrane area remain constant after hypotonic treatment? I.e. are the eisosomes really disassembled and not just diluted if the plasma membrane area increases?

Figure 4H: Looking at the time series images it seems like the plane of focus shifts during the experiment. At time zero, and at 1 minute, the focus is at the surface of the cell, but later it appears that the focus shifts toward the center of the cell. During the second half of the time series the remaining eisosomes are seen at the periphery of the cell, which could be explained by a focus shift. Is this a representative example?

Lines 325 : The authors could clarify a bit more clearly whether they quantify the density of endocytic events, or the density of Fim1 labeled endocytic sites, which would depend on the lifetime of the Fim1 patches. In other words, is the reported value the number of endocytic vesicles formed per area?

Line 340: I don't understand why the authors say that "volume increased faster than the change in endocytic density". Aren't they both significantly changed already at time 0, and it takes longer for the volume to reach maximum change?

Lines 379-382: This is the only place where potential non-vesicular mechanisms are brought up. Lipid transfer, for example, via membrane contact sites is a possible mechanism for expanding the plasma membrane area. The method used cannot distinguish between such non-vesicular transport and exocytosis. Therefore, the authors should bring this caveat up also in the discussion.

Lines 370-382: I think that the exocytosis assay is not totally intuitive. It might be useful for the readers if the authors tried to clarify the explanation a bit. In other words: how is it measuring exocytosis and not endocytosis, which FM4-64 is usually used to measure?

Line 520: How do the authors arrive at the value of 2%. Please elaborate.

Line 583: What is "higher order eukaryote"? (I would also advise against using "higher eukaryote", which, although commonly used, is not a clearly defined term, nor really biologically meaningful.)

---

## [Author Response]

[Editors’ note: the authors resubmitted a revised version of the paper for consideration. What follows is the authors’ response to the first round of review.]

Reviewer #1:This study is of interest as the role of eisosomes in fission yeast remains poorly explored and the findings presented by Lemière and Berro are likely to be relevant for the study of CME and its relation with membrane tension in mammalian cells. Several controls need however to be performed to better support the conclusions of this otherwise timely contribution.1. My main concern has to do with the readout used in this study. The authors assume that following the dynamics of fimbrin allow to monitor endocytosis and actin dynamics. What is the evidence that fimbrin molecules are recruited to endocytic patches? Several control experiments should be performed to validate this claim including clathrin colocalization to fimbrin patches. In addition, inhibiting CME (dyn mutant or else) should then result in decrease of fimbrin recruitment. Several of the effects of osmotic shocks described on fimbrin dynamics could occur independently from CME and the authors ought to correlate these effect with cargo uptake. What is the effect of fimbrin deletion on endocytosis? As for actin dynamics, why did not the authors follow actin directly? Following the dynamics of actin or another actin binding protein would reinforce this aspect of the work.

We thank Reviewer 1 for their interest in our study. We thank the reviewer for pointing out the lack of clarity about the marker we used to monitor actin dynamics. We have clarified our rationale in the main text. In summary, fimbrin is an actin filament crosslinker whose recruitment to CME structures has been well documented (Adams et al., 1991; Berro and Pollard, 2014; Drubin et al., 1988; Sirotkin et al., 2010; Sun et al., 2019). Fimbrin has been routinely used as an actin marker during CME because it is abundant and follows exactly actin dynamics during CME (it is recruited, peaks and disappears at the same time as actin). In addition, GFP-tagged fimbrin is fully functional. Fimbrin deletion in *S. pombe* has a very mild phenotype, probably because of compensatory effects from other crosslinkers or other proteins such as tropomyosin (Skau and Kovar, 2010). We did not use actin as a marker because GFP-actin is not fully functional, and one needs to overexpress GFP-actin on top of endogenous actin. Overexpression promoters in fission yeast lack precision and have significant cell-to-cell variability, which makes the data more noisy and difficult to quantitatively analyze and interpret. Colocalization of fimbrin with clathrin and adaptor proteins has been shown before (Kaksonen et al., 2005; Sirotkin et al., 2010). To make our argument more convincing, we have added in the paper a panel showing colocalization of fimbrin with the adaptor protein End4p, which is recruited before fimbrin and is a marker of endocytic structures and vesicles (Figure 1D and 1E). Note that we did not choose to use clathrin as a second marker because clathrin also localizes to endosomes in addition to the plasma membrane, which makes the signal more dispersed. It has been shown before that if early stages of endocytosis are inhibited, actin does not assemble at CME sites, neither does fimbrin (Sun et al., 2015). If actin dynamics is inhibited with drugs, endocytosis is blocked before any membrane ingression (Ayscough et al., 1997; Kukulski et al., 2012).

The successful formation of a vesicle can be assessed by the movements of the markers after they peak. Our new data using the FM4-64 dye (Figure 8) demonstrates that the dye is internalized even in the worst hypotonic conditions tested, which further demonstrates that vesicles are indeed formed. Figure 8B shows example images of internalized vesicles stained with the FM4-64 dye in wild-type walled cells. Author response image 1 is representative wild-type protoplasts stained with FM4-64 for 10 minutes after a hypotonic shock of -0.2 M (the movie from which this snapshot was extracted was used to produce Figure 8C).

2. I am somehow concerned by the degree of novelty of this study as it has long been known in mammalian cells that membrane tension controls the rate of endocytosis and exocytosis (see Sheetz seminal papers and more recently papers such as Boulant et al.,). Moreover, eisosomes have been recently established as membrane reservoirs in a study with similar data in yeast (protective role of the wall and disappearance of eisosomes through flattening under increased osmolarity in protoplasts as described in Kabeche, JCS 2015). As a matter of facts, these published data should be better acknowledged in the results and discussion. For instance, on line 178, the authors write that their data "suggest that eisosomes might be disassembled to reduce membrane tension in media with low osmolarity…" this has been already demonstrated in Kabeche, JCS 2015. In this context, the new finding reported in the current study is mostly about the role of endocytosis on membrane tension, that is, that endocytosis is transiently reduced to decrease membrane tension. However, in none of the presented experiments have the authors addressed the selective contribution of endocytosis. The authors should therefore re-examine their findings when endocytosis is blocked by drugs or mutants. Thus, measuring membrane tension variations under osmotic shock when endocytosis is blocked would allow to estimate the relative contribution of endocytosis in membrane tension buffering.

We thank Reviewer 1 for pointing out the lack of clarity in what was previously known and what is new in our study. We have edited the text to further describe what was already known before we performed our study and provided better reference to previous papers. We have also edited the text to better describe what are the novelties of our paper.

Some links between membrane tension, endocytosis and exocytosis have been described before, but our study brings a new quantitative characterization of these processes to determine (1) how much more actin is needed when membrane tension is increased, (2) how eisosomes influence this actin regulation, and (3) how much membrane is added via regulation of the rates of endocytosis and, in this revised version of the paper, via regulation of the rate of exocytosis. Kabeche et al., showed eisosomes disassemble under hypotonic shocks but did not prove that hypotonic shocks increased membrane tension or that membrane tension is reduced by eisosome disassembly. Our micropipette aspiration data, and our data on actin requirement during CME demonstrate that membrane tension is indeed reduced.

We thank the reviewers for suggesting us to determine the effects of blocking endocytosis on membrane tension. We have now included this experiment and an experiment where we blocked exocytosis (new Figure 9 in the revised paper). However, since direct membrane tension measurements with micropipette aspiration is difficult, low throughput and slow (it requires several minutes, which is the timeline over which the systems goes back to equilibrium), we do not think that the temporal resolution of membrane tension measurement using micropipette aspiration is high enough to monitor the effects of osmotic shocks or drug treatments. Instead, we used the fact that the plasma membrane ruptures and cells are more likely to die when membrane tension is high to determine the effects of blocking endocytosis or exocytosis (Figure 9 in the revised paper).

3. The absence of eisosomes should be documented in Pil1∆ mutants. Line 244: what is the evidence that WT protoplasts express normal amounts of Pil1p when they assemble fewer eisosomes? Data presented in Figure 6 should be correlated with the visualization of eisosomes as in Figure 4B.

We have clarified the text to better explain that previous studies have shown that the deletion of the gene coding for the major scaffolding eisosome protein Pil1p is sufficient to remove all eisosomes from the cell (Kabeche et al., 2011; Olivera-Couto et al., 2011).

Since wild-type protoplasts are made and imaged within 50 min, we did not expect any significant change in the expression level of Pil1p between intact cells and protoplasts. Indeed, we have now checked the concentrations and found no difference (Figure 4 – Supplement 1A). We have also added new panels H and I in Figure 4 to quantify the amount of eisosome that disassembles during a shock and the kinetics of the disassembly.

Reviewer #2:Lemiere et al., use quantitative fluorescence microscopy in the model organism fission yeast to address the complex relationship between cellular processes such as endocytosis and cellular structures such as eisosomes to regulate and equilibrate plasma membrane tension. Dissecting individual processes that mechano and/or biochemically regulate plasma membrane tension remains to be challenging. Single cellular organisms such as fission yeast therefore provide a versatile and easy to manipulate model system to tackle these particular very important regulatory processes and can translate crucial findings to higher eukaryotic model systems.The authors use their previously established quantitative fluorescence microscopy and analytical approach to measure the response of the actin crosslinker fimbrin to changes in the number of molecules, lifetimes and speed to osmolarity media changes that are meant to change plasma membrane tension. The authors describe in a combination of experiments conducted in walled cells and protoblasts, both also mutated for the Pil1 eisosome component, and changed dynamics of fimbrin. These findings provide very exciting new results about how the actin machinery adapts to osmolarity changes. These are even combined with membrane tension measurements. Together with determining the density of endocytic events after hypotonic shock treatment, the authors conclude that the cooperation of eisosome disassembly, decrease of endocytic events and ongoing exocytosis are responsible for equilibrating plasma membrane tension changes.The findings in the manuscript are of descriptive nature and do not identify mechno-regulatory or biochemical mechanisms that control the membrane reservoir or the adaption of actin dynamics in response to changing media osmolarities. Our major concern is that the overall main point of the paper is not clear. It seems that the authors present two very interesting half stories that have not yet have been completed and that do not stand on their own in a complete story. Given the allowed time frame of revision, it is not clear that the authors could make adequate revisions. If they can, we would suggest that they concentrate on one of their respective stories and endeavour to provide more mechanistic insights into either "the adaption of actin dynamics in yeast endocytosis in response to membrane tension changes" or "cellular mechanisms that control the plasma membrane reservoir in response to osmolarity changes". Detailed comments can be found below.1. As mentioned above, it seems that the authors pursued two different stories in this manuscript and tried to cover them under one main point. This creates confusion for the reader throughout the manuscript especially in Figure 4 and Figure 5.Therefore, we suggest that the authors concentrate on one of the two take subjects that they address. The two different stories that could be expanded for a manuscript are:a. One on the adaptive dynamics of the actin crosslinker fimbrin. The authors precisely report on the number of molecules, lifetimes and speed changes that happen under hypotonic shock treatments.However, the authors do not pursue how actin dynamics mechanistically adapt to hypotonic shock treatments. Fimbrin is a crosslinker and there are many other interesting candidate proteins whose dynamic behavior under hypotonic shock treatments the authors could explore. Thus, it would be possible for the authors to obtain precise mechanistic details about how "actin dynamics adapt to membrane tension increase to control yeast endocytosis". Previously published work of the last author shows that these strains are available. We would like to see experiments equivalent to the one conducted with the fimbrin-FP expressing yeast strain with Sla1, Arp2/3 and an early endocytic marker. If the authors are willing to add these experiments, we would also like to see the measurements of membrane tension of the protoblasts treated in the conditions used in Figure 5. A direct relationship between actin cytoskeleton protein binding dynamics and quantified membrane tension is of high interest for the endocytic community and would make this manuscript very valuable and significantly new. Then the authors could explain and discuss the differences they see in fimbrin dynamics, which they do not include on in the main text. In Figure 2E and Figure3D, fimbrin lifetimes are faster under hypotonic treatment. However, in Figure 5, lifetimes decrease under hypotonic treatment. We do not understand what the authors are looking at in these analyses. Determining membrane tension for all experimental conditions is essential for the correct interpretation of these results (of course, this is only possible for protoplasts). In addition, the authors should think about showing experimentally how faster or slower fimbroin dynamics can be related to productive endocytic events.

We thank the reviewers for finding our results exciting. The goal of our paper is to describe how the actin machinery is regulated after a sudden increase in membrane tension and how membrane tension rapidly adapts to change. We believe that the description of these phenomena and the relative contributions of the cell wall, eisosomes, endocytosis and exocytosis in this process will be useful for the community. We acknowledge that our paper does not identify all the underlying molecular mechanisms involved but lay the foundations to do so. It is in our plans to follow up on both parts of our study to determine the molecular mechanisms at the origin of the effects we observed. We believe that identifying molecular mechanisms for each part will each lead to a full paper and are therefore beyond the scope of our current study.

In our paper, we present two related effects of osmotic shocks because the data from the same experiments have been used to determine at the same time a) the amount of fimbrin assembled and b) the rate of endocytosis. Because these complementary results are extracted from the same datasets, we believe it is legitimate to present them in the same paper. In addition, the results of both parts are consistent with each other and presenting each part independently would make the results more puzzling and difficult to interpret.

To complete the story about membrane homeostasis, and to substantiate our speculations on the rates of exocytosis, we have adapted a method to determine the rates of exocytosis using the FM4-64 dye (new Figure 8 in the revised paper). We now show that (a) the rate of exocytosis is increased after a hypotonic shock in protoplasts but not in walled cells, and (b) change in endocytosis and exocytosis rates almost exactly account for the change of membrane surface area. We believe that these new data significantly complement and clarifies the second part of our paper.

Measuring membrane tension in different osmotic shocks so as to correlate the change in actin dynamics to membrane tension (instead of osmolarity) is an excellent suggestion. However, measuring membrane tension using micropipette aspiration is slow, slower than the membrane tension adaptation. Therefore, it would be challenging to accurately correlate the dynamics of the actin machinery with membrane tension. We and our collaborators recently obtained funding to build a new setup which includes optical tweezers and will allow us to make more precise measurements and also allow us to follow the temporal evolution of the adaptation of membrane tension. We believe data obtained with this setup.

b. The other main point the authors pursue is the regulatory mechanism for membrane tension equilibration in response to osmolarity changes. Here the authors suggest focus on these regulatory components: the endocytic machinery, eisosomes, the cell wall and the exocytic machinery. With their experiments the authors only cover the dynamic behavior of the actin crosslinker fimbrin as the representative marker for the dynamics of endocytic events in response to hypotonic treatment and the observation that PIL1 deleted cells are more prone to hypotonic shock.What follows are conclusions made by the authors despite a lack of experimental evidence:- 'eisosomes play a critical role in the regulation of membrane tension and membrane integrity through dynamic remodeling and scaffolding of the plasma membrane' p9, line 342.- the 'protective role of eisosomes is even more striking in protoplasts under acute hypotonic shocks'.- 'our data indicate that membrane tension is decreased via the disassembly of eisosomes, through release of excess membrane surface area.' p9, line 352.- Lines 361 – 367: 'Our study identified a third, new mechanism to reduce membrane tension by temporarily reducing the endocytic density. Using our data and previous measurements (18,19), we estimate that cells endocytose about 2% of their surface area per minute. Assuming that at steady state the surface area added to the plasma membrane via exocytosis balances the surface area that is endocytosed, a reduction of the endocytic rate while keeping the exocytic rate constant for a few minutes would allow for a net addition of surface area to the plasma membrane'.The authors only show images of eisosomes at steady state e.g. Figure 4a,b,c.To investigate the regulatory mechanism and provide convincing evidence for their claims, we suggest that the authors also track Pil1 dynamics under different hypotonic treatments and the dynamics of an exocytic marker. Here again, a direct relation to membrane tension values measured under the experimental conditions used would be crucial.

We thank the reviewer for pointing out the ambiguity of these statements or their overinterpretation in the initial version of our paper. We have modified them to make sure credit is given to the previous studies that support these statements and how our data support these claims (originally l. 342 and 352). In addition, the new version of the paper contains data about the rates of exocytosis which support the last statement mentioned above (originally l. 361-367). We have also added new data showing the temporal evolution of eisosomes at the plasma membrane after a hypotonic shock (Figure 4H and 4I).

2. The title of the manuscript, "Adaption of actin dynamics and membrane tension control for yeast endocytosis," does not describe the work well since the authors only perform experiments on the actin crosslinker fimbrin and quantification of membrane tension was only performed for one experimental condition. We suggest that the authors choose a more specific title that states the main point of the revised manuscript.

We have modified the title to better reflect the main message of the revised paper.

3. "Temporal super-resolution" is not yet a common method that the reader should be expected to know about. Moreover, the reader might not be familiar with the improvements of this method performed by the authors (Line 98). We suggest that they schematically describe the method of temporal super-resolution in a supplementary Figure to Figure 1 and also explain the improvements. It is also not clear in which plane the cells are imaged and why only a subset of fimbrin tracks where kept for the further analysis (see comment 4.)

We have extended the description of the temporal super-resolution method in the

“Measurement of the temporal evolution of the number of proteins and speed” section of the Materials and methods. We have also made sure that we clearly explain that we imaged the “bottom” surface of the cell, i.e. the planes closest to the objective and the coverslip.

4. It would be helpful if the authors would provide a thorough analysis of how many fimbrin tracks there are in total, what classes of behavior can be seen and on which subset the authors are focusing and why.The authors state in their methods, that all tracks were discarded that did not match the following criterion: 'longer than 5s and displaying an increase followed by a decrease in intensity' p12 line 497. This introduces a significant bias towards productive tracks. Tracks that are stalled or last longer than the observation window would not enter the analysis.Why did the authors did not discard tracks that 'failed to proceed normally, as reported by the virtually null speed' p6 line 196? Or did they use a different criterion for these tracks?

We thank the reviewers for pointing out the ambiguities in our description of the methods. While developing our analysis pipeline, we were careful not to introduce bias when we chose the methods and parameters for tracking and selecting the tracks to keep. Sites of endocytosis are quite dense in most areas of the cell, which means that a very large majority of the fluorescence signal from endocytic events overlap with each other and cannot be used for quantitative analysis. It is difficult for tracking algorithm to identify overlapping spots, independent spots that appear consecutively in a diffraction limited region, or spots that go out of focus, etc. Our methods aimed at rejecting those spots automatically when possible, or manually after visual inspection. In fact, after visual inspection, long tracks always corresponded to multiple endocytic events that overlapped (which can usually be determined by the shape of the fluorescence spot or a sequence of multiple fluorescence increase and decrease), and tracks shorter than 5 s never covered the full time course of an endocytic event.

Speed was not one of our criteria because overlapping patches cannot be distinguished by speed alone. Therefore, we did not reject tracks that did not move much and noticed that some tracks had very low mobility in some conditions (e.g. l. 196 of the original paper as Reviewer 2 noted).

We do not believe there are several classes of endocytic events, because even when tracked manually or automatically but with very lenient parameters, we identified only a single class of behavior for single endocytic sites.

5. Quality of Figures: Overall the plot quality could be improved by reporting the confidence interval differently. The plots become very busy and the lines overlap strongly, so that the reader has to look at the plots with a higher zoom to differentiate between the actual data lines and the lines of the confidence interval. We suggest that they plot the lifetimes of endocytic events separately and report statistical tests on the lifetime distributions. The way the Figure legends are placed in Figure 4 is very confusing and can be improved in the other figures for clarity, as well.

We thank Reviewer 2 for these suggestions. To increase clarity, we have added new supplemental figures where we have plotted the curves individually for different conditions whenever they significantly overlapped in the main figures. We have also performed statistical tests when it was relevant.

Patch lifetimes are virtually impossible to determine accurately since our tracking method detects spots only when the signal to noise ratio is high enough. Therefore, we likely miss at least one or two time points at the beginning and the end of each track. Determining lifetimes with high precision would require us to extrapolate the data, which we prefer not to do. In the paper, we discuss relative differences in lifetimes by comparing the times when the number of molecules is half the maximum value (which is reached at time 0 s). Since this number is evaluated from the mean number of molecules we have only one data point per condition and we cannot perform statistical analysis between conditions. To highlight the lack of precision in these numbers, we only discuss the difference between conditions as a percentage, round these numbers to the closest multiple of 5 and added an approximate sign (~) before them.

6. A model Figure summarizing their main conclusions at the end of the manuscript would clarify the take home message for the reader. The summary could make points about: What factors influence membrane tension control? Or, they could provide a schematic representation for how the actin machinery adapts to increases in membrane tension.

We thank Reviewer 2 for this suggestion. We have added a new figure at the end of the paper with a schematic summarizing our results (Figure 10).

7. The plots in Figure 5H and I and the conclusion in the text about the Pil1 cytoplasmatic pool are very confusing and would be better placed in a supplementary Figure.

The rationale for showing these figures in the main text is to demonstrate that once we mechanically get rid of most of the eisosomes (by putting protoplasts in a solution with a very low osmolarity), endocytosis behaves the same way as in cells where the eisosome is genetically removed. We believe this further validates our conclusion that eisosomes are the first line of defense from a mechanical shock. We think these figures are an important piece of data to make our point so we have left them in the main figures. However, we have edited the text to make these ideas clearer.

Reviewer #3:The paper by Lemiere and Berro focuses on the dynamics of endocytic events and eisosomes in walled cells and protoplasts, to investigate the role of membrane tension in both endocytic bud formation and eisosomes. It builds on previous work that has shown that clathrin endocytosis dynamics is affected by changes in membrane tension, and that eisosomes participate into the regulation of plasma membrane tension. The paper is clearly written, and has interesting findings, among which are:1. That membrane tension changes do not strongly affect CME in walled cells2. Membrane tension strongly affect CME in protoplasts3. Eisosomes, while having a limited role in walled cells, have a strong role in regulating membrane tension in protoplasts.I have a few concerns about the paper in the current state. The authors want to compare WT (walled) cells and protoplasts, but the comparison is not done thoroughly, which makes the paper a bit unbalanced. Also, the fact that the paper is written in two parts, first the walled cell results, and then the protoplasts results, the comparison is not very easy to follow. The authors may reconsider the text organization by clustering similar experiments done in wall cells and protoplasts. I understand that some experiments cannot be performed in protoplasts or in walled cells, but the following comments are here to extend as much as possible the comparison.

We thank Reviewer 3 for finding our findings interesting. As we explained in our response to other reviewers, we have kept both parts together because most of the results from both parts come from the same experimental datasets, and the results of both parts support each other. We have extended the membrane tension and homeostasis regulation parts with new data on the rates of exocytosis. We have considered reorganizing the text around mutants but we think our message is still clearer when the results on the dynamics of the actin machinery and on membrane tension regulation are separated. We believe this organization is also better justified and clearer now that we have added new data about the regulation of the rates of exocytosis (Figure 8).

– The density of eisosomes should be better characterized in walled cells and in protoplasts, before and after the shock (see Figure 4B and 4C, if I understood well, this is only for cells at steady state, and different osmotic pressure). The authors model proposes that eisosomes regulate tension through disassembly. The authors have to show this. I would recommend to also perform hypertonic shocks that should increase the number of eisosomes.

We thank Reviewer 3 for this suggestion. We have added new live imaging data of Pil1p that allowed us to quantify the timing and extent of eisosome disassembly after an acute hypotonic shock (Figure 4H and 4I in the resubmitted manuscript). These data strengthen our argument that eisosome disassembly regulate membrane tension.

We agree with Reviewer 3 that understanding the dynamics of eisosomes under hypertonic shocks is an interesting and important question. However, we believe answering this question is beyond the scope of our current paper, which focuses only on increases in membrane tension (i.e. under hypotonic shocks). There are several pieces of evidence in the literature that argue that the response to hypertonic shocks triggers different pathways than the response to hypotonic shocks. To keep our paper clearer we have decided to leave the study of hypertonic shocks on eisosomes for a future study.

– Generally speaking, it sounds strange that eisosomes would have an impact only on protoplast. The observation that when grown at higher osmotic pressure, cells have more eisosomes suggest that cells prepare to stringer hypo-osmotic shocks in this condition. How the author can rationalize these results?

Our initial data showed that eisosomes did not seem to have a significant effect on walled cells after a single hypotonic shock while had a protective effect in protoplasts. However, while performing experiments for the revisions of this paper, we serendipitously discovered that eisosomes do protect walled cells from dying when we performed consecutive osmotic shocks (Figure 6D and 6E). We have now revised our conclusions to reflect the protective role of eisosome more accurately.

– I was wondering how sure the Fim1p reporter reports actin density? I guess this has been published elsewhere as the authors rely on published work on this point, but it would still be nice to have a figure panel that shows a co-labelling with actin.

As we explained in our answers to other reviewers, fimbrin is a marker for actin dynamics that we and other have used extensively in endocytosis because it is very abundant and exactly follows the dynamics of actin in wild-type and all mutants tested. To clarify this point we have added new panels showing colocalization of fimbrin and the clathrin-coated pit adaptor End4p (homolog of Sla2 in *S. cerevisiae* and Hip1R in mammals) (Figure 1D and 1E).

– I find surprising that the dynamics of actin is not different upon osmotic shocks in WT walled cells. I was wondering if the actin treadmilling was modified upon osmotic shocks. This should be easy to test through FRAP experiment of Fim1p at the endocytic sites.

We say in the text that the overall dynamics of actin does not change after an osmotic shock. However, the microscopy method we used cannot tell whether the turnover of actin dynamics is different in these conditions. This would be an interesting property to measure. However, performing and analyzing FRAP data on such transient system is very difficult. We recently collected some FRAP data for another paper (Lacy et al., 2019) but we realized that even at steady-state (i.e. not in shock conditions) the resolution of the data would not be high enough for us to make definitive conclusions.

– Reported tension values are extremely small (10-4 nM/um, which is 10-7 N/m). Fibroblastic cells have tensions of typically 10-4 – 10-5 N/m. I expect yeast cells to have even higher tension become they live in higher turgor pressure. Also at 10-7 N/m, membrane usually have large fluctuations which are visible by any type of microscopy, which are not seen in the authors's experiments. Either the calculation has a problem, or the unit is wrong. Please use standard units for membrane tension in N/m or in mN/m (or m-1).

We thank Reviewer 3 for picking up this typo. Indeed, the units we meant to report are actually 10^3^ larger than initially reported on the figure. Our numbers are indeed consistent with the values mentioned in the reviewer’s comment and the values previously reported in the literature.

[Editors’ note: what follows is the authors’ response to the second round of review.]

Reviewer #1:This submission is a revised version of a previous submission. The authors have done a substantial amount of work, and the paper has increased in length. Some aspects have greatly improved, such as the characterization of the roles of the cell wall, tension and actin (using drugs and mutants to change actin organization) in controlling endocytosis. The role of eisosomes in buffering membrane tension has been characterized in further details, providing the first evidence that eisosomes could play a similar role in yeast than caveolae in cells in buffering membrane tension. Finally, the authors added a study on the evolution of exocytosis during hypoosmotic shocks, which reacts in opposite direction than endocytosis.While I value the efforts and work provided by the authors in their revised version, they in fact took an opposite direction in their revised manuscript than what was initially suggested by the reviewers: – split the manuscript in two, and strengthen the characterization of tension/endocytosis on the one hand, and eisosomes/tension on the other. Instead, the authors kept the two parts in the same manuscript, and added a third part on exocytosis. They also extended their comparison of walled cells and protoplasts, which is indeed interesting, but makes the flow hard to follow because the reader needs to go back-and-forth between different figures. This choice is at the expense of the manuscript's readability, which became lengthy and difficult to follow.While I value the efforts made by the authors, and the improvement of the characterization of endocytosis, tension and eisosomes, I am embarrassed to say that the main criticism remains. I understand that the authors wanted to broaden their study, and get more general conclusions, but it does not go through well. I unfortunately cannot recommend publication in the present form, and would certainly suggest again to split the manuscript in two (which could be back-to-back in eLife for example), one on endocytosis/exocytosis and tension (where pil1delta and protoplasts would be used as a tool to dysregulate tension) and one on eisosomes/tension buffering.

We thank Reviewer #1 for her/his positive comments about the new work presented in this revised version. We thank her/him for pointing out to us that the paper is still difficult to read, even more so after the revisions. On their suggestion, we have thoroughly reorganized the paper into two parts, the first one addressing the regulations of membrane tension by eisosomes, endocytosis and exocytosis (Figures 1 to 5) and a second part on the effects of increased membrane tension on the dynamics of actin at sites of endocytosis, focusing on wild-type and *pil1Δ* protoplasts as tools to amplify the increase in membrane tension (Figure 6 and 7). We thank the reviewer for this suggestion because we think this new organization indeed makes the flow of the paper better and its messages clearer. We seriously considered splitting our study into two papers, but we eventually decided to keep everything in a single paper because we believe the data we present are quite interconnected and splitting the paper into two would create other readability issues and may force readers to go back and forth between papers. That said, if Reviewer #1 still finds our new version difficult to read we should be able to split it in two papers.

– While the mechanism at play to regulate dynamics of endocytosis and eisosomes during osmotic shock became well characterized, the new part on exocytosis is very short and does not provide any further understanding of the regulation at play here. Is it tethering, or SNARE function that is regulated by tension? Is that higher tension strengthens the actin dome around exocytic vesicles to trigger fusion? A better characterization, at least at the same level of other parts would be needed here.

Our new data quantitatively describe how much more membrane is added by exocytosis to regulate membrane tension. Understanding the precise molecular mechanisms at play during the regulations of exocytosis when membrane tension increases is indeed an important and interesting question. However, it is not an easy question to address and there have been debates about the effect of high membrane tension on exocytosis (Kozlov and Chernomordik, 2015). In particular, two contradictory effects happen at the same time: a) the initiation of the fusion of a lipid vesicle to a lipid bilayer is less favorable under high tension, but b) once fusion starts, high membrane tension favors the completion of the fusion. In addition, tethering complexes (like the exocyst) and the SNARE complex may also be influenced differentially by membrane tension. In other words, the question is complex and it would not be straightforward to answer it in vivo, especially over the two months allocated for the revisions. Note we can already exclude that actin plays a direct role in favoring vesicle fusion in yeast cells because, unlike mammalian cells, they do not have an acto-myosin cortex, and the only actin at the plasma membrane is at site of endocytosis. The only indirect role of actin in exocytosis may be in vesicle delivery to the plasma membrane via actin cables.

– Some plots are difficult to read. The plot "Speed" vs "time", which zero the invagination time, is not easy to read. The total duration of endocytic events is changing with conditions, and if I understood well, that is the main effect that the author look at, but it took me quite some time to figure out how to read these. I think a classic plot with average duration would be easier to follow for most readers. I understand that in some cases, the authors see no invagination, which is also reported in this graph, but I think 2 plots would do a better job than a single one in this case. Or perhaps, explain better how to read those plots, as they are uncommon, and at the basis of the authors' argumentation.

The “speed vs time” plots allow us to determine whether endocytosis completes normally in different conditions. At a given time, the speed corresponds to the average movement of the endocytic structure in the plane of the membrane (XY plane) between two consecutive images. For an endocytic event that completes normally, the speed is close to 0 in the XY plane while the endocytic pit elongates along the Z axis before the vesicle is pinched off. Note that the speed is not exactly 0 because of localization errors and putative small movements of the endocytic structure in the plane of the membrane. The speed increases after the vesicle is pinched off and it diffuses freely in the cytoplasm. Since this movement is mostly diffusive, the standard deviations of the speeds are large. To clarify the text, we have added an explanation on how to read these figures in the Materials and methods.

Note that the durations of endocytic events in different conditions can be compared directly from the “number of molecule vs time” plots, but they cannot be precisely determined because fluorescence intensities are too low at the very beginning and very end of each endocytic event to be detected unambiguously, which prevents us to determine the exact time for appearance and disappearance.

– I disagree with the use of deltaP to account for a change of osmolarity. Pressure should be in Pa (or any corresponding unit), not in Moles per Liter. Either the authors use deltaC, or they measure the osmotic pressure of their solution and can use Osm in this case for pressure. Also, at concentrations about 0.5M, there is usually a substantial discrepancy between Osmolarity and Molarity, and as the authors are using Molarities above 1M, I think it would be safer to estimate the osmotic pressure of those solutions.

We have changed our nomenclature to express molarity changes as ΔC instead of ΔP.

Reviewer #2:Lemiere and colleagues show in this manuscript that endocytosis is remarkably robust against osmotic changes in normal walled yeast cells. However, in cells without cell walls endocytosis is much more sensitive to osmotic changes, presumably due to changing membrane tension. Eisosomes are shown to be important for regulating membrane tension after osmotic shocks.Changes in turgor pressure are also shown to transiently affect the frequency of endocytic events. Eisosomes are shown to be important for restoring the normal endocytic frequency after shocks. Finally, exocytosis is suggested to be a mechanism that restores normal membrane tension after hypotonic shocks.The strength of the manuscript is the highly quantitative and careful experimentation and the fact that three different plasma membrane processes are considered together (endocytosis, exocytosis, eisosomes), which is surprisingly rarely done, even though they must be functionally connected.

We thank Reviewer #2 for their positive remarks about our revised manuscript.

I have some suggestions for the authors to improve their manuscript:Lines 217-223: Does the plasma membrane area remain constant after hypotonic treatment? I.e. are the eisosomes really disassembled and not just diluted if the plasma membrane area increases?

The plasma membrane area increased after hypotonic shock, but the normalized area fraction of eisosomes (total area of eisosomes divided by total area of protoplast, in units of um^2^) decreased as quantified in Figure 1I. Therefore, our conclusion is that eisosomes disassembled rather than being just diluted. The disassembly of eisosomes can also be observed in the representative images in Figure 1H (compare the second and the third frame), where the drop in eisosome coverage in the middle of the protoplast is stronger than the increase in protoplast surface area.

Figure 4H: Looking at the time series images it seems like the plane of focus shifts during the experiment. At time zero, and at 1 minute, the focus is at the surface of the cell, but later it appears that the focus shifts toward the center of the cell. During the second half of the time series the remaining eisosomes are seen at the periphery of the cell, which could be explained by a focus shift. Is this a representative example?

Because Figure 1H (originally Figure 4H) is a projected image, eisosomes at the periphery appear darker because fluorescence intensities are summed over several Z-slices whereas only one slice contributes to the fluorescence in the center of the projected image. From our experience with this setup, the focus is usually quite stable for at least 30 minutes.

Lines 325: The authors could clarify a bit more clearly whether they quantify the density of endocytic events, or the density of Fim1 labeled endocytic sites, which would depend on the lifetime of the Fim1 patches. In other words, is the reported value the number of endocytic vesicles formed per area?

The reported values are indeed the number of endocytic vesicles normalized to the cell length (we normalize to the cell length, not the cell area because cell area is more difficult to measure accurately due to the cell shapes). The fact that the lifetime of Fim1p may be different in different condition is accounted for by the method (we recalculate the temporal average of Fim1p at one endocytic site for each condition). To clarify this point, and because the counting method is not commonly used, we added a short description of the method in the main text.

Line 340: I don't understand why the authors say that "volume increased faster than the change in endocytic density". Aren't they both significantly changed already at time 0, and it takes longer for the volume to reach maximum change?

We clarified this sentence as follows:

“Note that the change in cell volume (Figures 3A and 3B, insets) could not exclusively account for the observed decrease in the endocytic density in wild-type cells as the relative increase in cell volumes were larger than the relative decrease in endocytic density 2 minutes after the shocks and remained large 4 minutes after while the endocytic densities recovered their pre-shock values.”

Lines 379-382: This is the only place where potential non-vesicular mechanisms are brought up. Lipid transfer, for example, via membrane contact sites is a possible mechanism for expanding the plasma membrane area. The method used cannot distinguish between such non-vesicular transport and exocytosis. Therefore, the authors should bring this caveat up also in the discussion.

We now discuss these alternative non-vesicular mechanisms also in the discussion and added the following paragraph:

“For the calculations presented above, we have assumed that the size of endocytic and exocytic vesicles remain constant in all osmotic conditions. However, further experiments will be necessary to validate or invalidate this assumption. In addition, our FM4-64 data does not allow us to distinguish between lipid addition to the plasma membrane via exocytosis or via a putative transfer of lipids by non-exocytic mechanisms (Reinisch and Prinz, 2021) and for simplicity and by lack of further evidence, we have discussed our data as an increase in the exocytosis rate. Not much is known about lipid transfer proteins at the plasma membrane and further experiments will be necessary to determine whether the activity of these proteins is enhanced by abrupt increases in membrane tension or hypotonic shocks.”

Lines 370-382:I think that the exocytosis assay is not totally intuitive. It might be useful for the readers if the authors tried clarify the explanation a bit. In other words: how is it measuring exocytosis and not endocytosis, which FM4-64 is usually used to measure?

We have modified the main text and provided references where similar approaches were used to better explain our assay. The paragraph now reads:

“To measure the rate of exocytosis in different conditions, we used the cell impermeable styryl dye FM4-64, whose fluorescence dramatically increases when it binds to membranes (Cochilla et al., 1999; Gachet and Hyams, 2005; Richards et al., 2000). After FM4-64 is introduced to the media, the plasma membrane is rapidly stained (Figure 4A). Fusion of unstained intracellular vesicles to the plasma membrane results in an increase of total cell fluorescence, because after each fusion event new unstained membrane from the interior of the cell is exposed to the dye. At this stage, if one wanted to measure endocytosis rates, one would typically remove the FM4-64 dye from the media to destain the plasma membrane, and quantify the internal fluorescence which is proportional to the amount of membrane internalized by endocytosis. Here, we kept the FM4-64 dye in the media, and monitored the fluorescence of the plasma membrane and the interior of the cell. In this case, endocytic events do not increase the total cell fluorescence because they transfer patches of the plasma membrane that are already stained into the interior of the cell (Figure 4A). Therefore, the increase in fluorescence we measured is due to the addition of new unstained lipids to the plasma membrane by exocytosis.”

Line 520: How do the authors arrive at the value of 2%. Please elaborate.

To clarify our calculation, we added the following paragraph in the Materials and methods:

“We measured ~120 endocytic sites per cell at a given time on average. Since the actin meshwork assembles and disassembles in about 20 seconds, we estimate that 360 endocytic vesicles are formed per minute. Assuming endocytic vesicles have a 50-nm diameter, this corresponds to a ~2.8 μm2 endocytosed per minute, or ~2% of the total surface area, considering the average cell is around 12 μm long and has a 132 μm2 of plasma membrane (assuming the cell shape is a cylinder capped by two hemispheres).”

Line 583: What is "higher order eukaryote"? (I would also advise against using "higher eukaryote", which, although commonly used, is not a clearly defined term, nor really biologically meaningful.)

We have corrected this typo to “other eukaryotes”.